# A phase transition enhances the catalytic activity of SARM1, an NAD+ glycohydrolase involved in neurodegeneration

Heather S Loring[1,2], Victoria L Czech[3], Janneke D Icso[1,2], Lauren O'Connor[3], Sangram S Parelkar[1,2], Alexandra B Byrne[3], Paul R Thompson[1,2]*

[1]Department of Biochemistry and Molecular Pharmacology, UMass Medical School, Worcester, United States; [2]Program in Chemical Biology, UMass Medical School, Worcester, United States; [3]Department of Neurobiology, UMass Medical School, Worcester, United States

**Abstract** Sterile alpha and toll/interleukin receptor (TIR) motif–containing protein 1 (SARM1) is a neuronally expressed NAD+ glycohydrolase whose activity is increased in response to stress. NAD+ depletion triggers axonal degeneration, which is a characteristic feature of neurological diseases. Notably, loss of SARM1 is protective in murine models of peripheral neuropathy and traumatic brain injury. Herein, we report that citrate induces a phase transition that enhances SARM1 activity by ~2000-fold. This phase transition can be disrupted by mutating a residue involved in multimerization, G601P. This mutation also disrupts puncta formation in cells. We further show that citrate induces axonal degeneration in *C. elegans* that is dependent on the *C. elegans* orthologue of SARM1 (TIR–1). Notably, citrate induces the formation of larger puncta indicating that TIR–1/SARM1 multimerization is essential for degeneration in vivo. These findings provide critical insights into SARM1 biology with important implications for the discovery of novel SARM1-targeted therapeutics.

*For correspondence:
Paul.Thompson@umassmed.edu

**Competing interests:** The authors declare that no competing interests exist.

## Introduction

Despite diverse clinical manifestations, axonal degeneration is an underlying feature of traumatic brain injury, peripheral neuropathies, and other neurodegenerative diseases, including Alzheimer's disease, Huntington's disease, and Parkinson's disease. These diseases account for extensive morbidity and mortality as there are no approved therapies. SARM1 has recently emerged as a promising therapeutic target for diseases associated with axonal degeneration because SARM1 knockdown or knockout prevents axonal degeneration and disease pathology (*Geisler et al., 2016*; *Henninger et al., 2016*; *Loring and Thompson, 2020*).

SARM1 was originally identified in a screen that selected for mutants that suppress injury–induced axonal degeneration (*Osterloh et al., 2012*). However, it was unknown how SARM1 loss prevented neurodegeneration post–injury. Subsequently, SARM1 was shown to possess enzymatic activity and catalyze the hydrolysis of NAD+ (*Figure 1A*; *Essuman et al., 2017*). Depletion of the NAD+ pool promotes degeneration, whereas inactive variants of SARM1 or supplementation with NAD+ precursors and minimizing NMN accumulation prevent degeneration after injury (*Essuman et al., 2017*; *Gerdts et al., 2015*; *Wang et al., 2015*).

SARM1 consists of three domains: a HEAT/armadillo (ARM) domain, two tandem sterile alpha motif (SAM) domains, and a C–terminal toll interleukin receptor (TIR) domain that catalyzes NAD+ hydrolysis (*Figure 1B*). The TIR domain contains a BB loop (Residues 594–605), which is thought to

**Figure 1.** SARM1 reaction and domain structure. (**A**) SARM1-mediated hydrolysis of NAD$^+$ results in the production of nicotinamide and a mixture of ADPR and cADPR. (**B**) Domain architecture of SARM1. SARM1 consists of three domains: an autoinhibitory HEAT/armadillo (ARM) domain, two tandem sterile alpha motif (SAM) domains that promote octamerization, and a C–terminal toll interleukin receptor (TIR) domain that catalyzes NAD$^+$ hydrolysis.

promote self–association, and a SARM1–specific (SS) loop (Residues 622–635), which is a characteristic feature of catalytically active TIR domains; TIR domains lacking this loop do not possess enzymatic activity (*Carlsson et al., 2016*; *Horsefield et al., 2019*; *Loring and Thompson, 2020*; *Summers et al., 2016*). Additionally, E642 (within the TIR domain and adjacent to the SS loop) is thought to be a key catalytic residue based on structural alignments with other nucleotide hydrolases and mutagenesis studies (*Essuman et al., 2017*; *Gerdts et al., 2015*; *Gerdts et al., 2013*; *Loring and Thompson, 2020*). In the absence of disease, the ARM domain is thought to inhibit catalysis via an intramolecular interaction with the TIR domain as deletion of this domain results in constitutively active SARM1 (*Chuang and Bargmann, 2005*; *Gerdts et al., 2013*). This hypothesis is supported by recent cryoEM structures of the inactive enzyme, which show that SARM1 exists as an octamer with the ARM domain binding to the TIR domain (*Bratkowski et al., 2020*; *Jiang et al., 2020*; *Sporny et al., 2020*). Mutations that disrupt the TIR–ARM interface or delete the ARM domain entirely, however, only activate SARM1 by ~1.8- to 2.4-fold (*Jiang et al., 2020*). Thus, the fully active conformation/state of the TIR domains remains to be determined.

Although SARM1 inhibitors hold therapeutic promise, efforts to study the purified enzyme have been complicated by the fact that SARM1 appears to lose activity during purification and only shows measurable NAD$^+$ hydrolase activity when assayed at a final enzyme concentration of $\geq$ 20 µM (*Horsefield et al., 2019*; *Loring et al., 2020a*). By contrast, enzymatic activity is 800-fold higher in lysates or when activity is measured on–bead (*Essuman et al., 2017*; *Horsefield et al., 2019*; *Loring et al., 2020a*). These data indicate that the pure protein does not fully recapitulate all features of SARM1 catalysis. Herein, we report that macroviscogens induce a phase transition that increases in vitro SARM1 activity by >2000-fold. By exploiting this finding, we report the first detailed studies of SARM1 catalysis. Additionally, we demonstrate that a point mutation which reduces multimerization and precipitation in vitro, also disrupts this phase transition in cell culture. We further show that citrate induces axonal degeneration in *C. elegans* that is dependent on the *C. elegans* orthologue of SARM1 (i.e. TIR–1). Notably, citrate induces the formation of larger puncta indicating that TIR–1/SARM1 undergoes a phase transition that is essential for degeneration in vivo. In total, these findings will aid efforts to therapeutically target SARM1 to treat neurodegenerative diseases as they improve our understanding of SARM1 activation and its role as a critical switch in the degenerative process resulting in catastrophic axonal fragmentation.

## Results and discussion

We previously established that SARM1 loses activity during purification and is essentially inactive unless purified to high micromolar concentrations (*Figure 2A*; *Loring et al., 2020a*). We initially hypothesized that the lack of activity could be explained by the loss of an essential element during enzyme purification. However, the addition of purified SARM1 to cell lysates did not activate the enzyme (*Figure 2B*). We then hypothesized that pure SARM1 adopts an inactive conformation. To investigate this hypothesis, we evaluated the effect of enzyme concentration (5–35 µM) on the rates of catalysis. Notably, the concentration dependence has parabolic character, increasing with upward curvature, as opposed to linearly, suggesting the formation of a more active oligomer (*Figure 2C*). The fact that pure SARM1 regains activity upon concentration, is inconsistent with the loss of an essential cofactor during purification. To investigate the oligomerization status of the SARM1 TIR domain, we next performed analytical ultracentrifugation (AUC). SARM1 was largely monomeric (~90%), as indicated by an S-value of 1.9–2.0S (*Figure 2D*). A smaller peak at ~3.9–4.2S, which showed rightward tailing indicates that the remaining ~10% exists as a trimer or tetramer (~71–88 kDa) with a smaller fraction in higher ordered forms. Similar oligomerization levels were observed at the two different SARM1 concentrations tested (*Figure 2D*).

Having shown that SARM1 activity increases nonlinearly with concentration, we next hypothesized that molecular crowding might activate the enzyme. To test this hypothesis, we measured the activity of SARM1 in the presence of several macroviscogens (i.e. PEG 8000, 3350, 1500, 400, and Dextran) and microviscogens (i.e. Sucrose and Glycerol). Microviscogens are often used as protein stabilizers and affect the ability of small molecules to diffuse in solution, whereas macroviscogens alter the free volume available for a molecule to move within a solution with little effect on diffusion (*Blacklow et al., 1988*; *Gadda and Sobrado, 2018*; *Hagen, 2010*). Therefore, microviscogens can reveal effects on diffusional processes, whereas macroviscogens affect molecular crowding (*Gadda and Sobrado, 2018*). Notably, we found that all the macroviscogens tested increased activity, whereas the microviscogens had negligible effects (*Figure 2E*). Specifically, PEGs 8000, 3350, 1500, and 400 increased activity by 1500, 2000, 1900, and 200–fold, respectively. By contrast, dextran (36–50 kDa) only increased activity by 14–fold (*Figure 2E*). To better visualize the effect of PEG 3350 on the rate of the reaction, representative progress curves are shown at 20 µM SARM1 with and without PEG (*Figure 2F*). The 20 µM reaction with PEG 3350 plateaus after ~ 350 s, whereas the fluorescence of the reaction without PEG increases at a much slower rate. In total, these data indicate that a subset of macroviscogens dramatically increase SARM1 activity by up to 2000–fold.

To determine how macroviscogens increase SARM1 catalysis, the activity of SARM1 (10 µM final) was next measured at 0, 10, 20, and 30% w/v of the macroviscogens and glycerol. Consistent with our hypothesis, we observed a dose–dependent increase in activity for all the macroviscogens tested (*Figure 2G*). Notably, activity increases according to macroviscogen size until a critical point is reached where PEG 1500 and PEG 3350 have similar effects (*Figure 2G*). Note that the effect on activity is not purely due to a change in viscosity, as the enhancement in activity shows a poor correlation ($R^2$=0.46) with viscosity (*Figure 2H*). For example, when dextran and PEG 3350 are assayed at similar relative viscosities, the fold activation is dramatically different (*Figure 2H–I*). Additionally, when the viscosity of glycerol was compared to that of PEG 3350, we found that glycerol did not enhance activity to the same degree as PEG 3350, despite having a similar relative viscosity (*Figure 2—figure supplement 1A–B*).

As PEG 3350 showed the greatest ability to enhance activity, we proceeded to investigate its effect on SARM1 activity further. Initially, we evaluated the effect of enzyme concentration on activity in the presence and absence of PEG 3350 (*Figure 2J*). SARM1 concentration was varied from 300 nM to 20 µM with 25% w/v PEG 3350 and from 5 to 35 µM in the absence of PEG 3350. Our data show that the addition of 25% w/v PEG 3350 accelerates the rate of reaction 7700-fold at 5 µM, 1400-fold at 10 µM, and ~280-fold at 20 µM of SARM1 (*Figure 2J*). These data indicate that the effect of PEG 3350 is dependent on enzyme concentration. Furthermore, SARM1 activity appears to increase almost linearly with respect to concentration in the presence of PEG after a slight lag phase, but parabolically in its absence (*Figure 2J*). This trend suggests that in the absence of PEG, there is a gradual build up in activity as the concentration of SARM1 increases, until the concentration surpasses a critical threshold allowing for increased transient interactions and a consequent large increase in activity (*Figure 2J*). By contrast, the addition of PEG reduces the concentration of

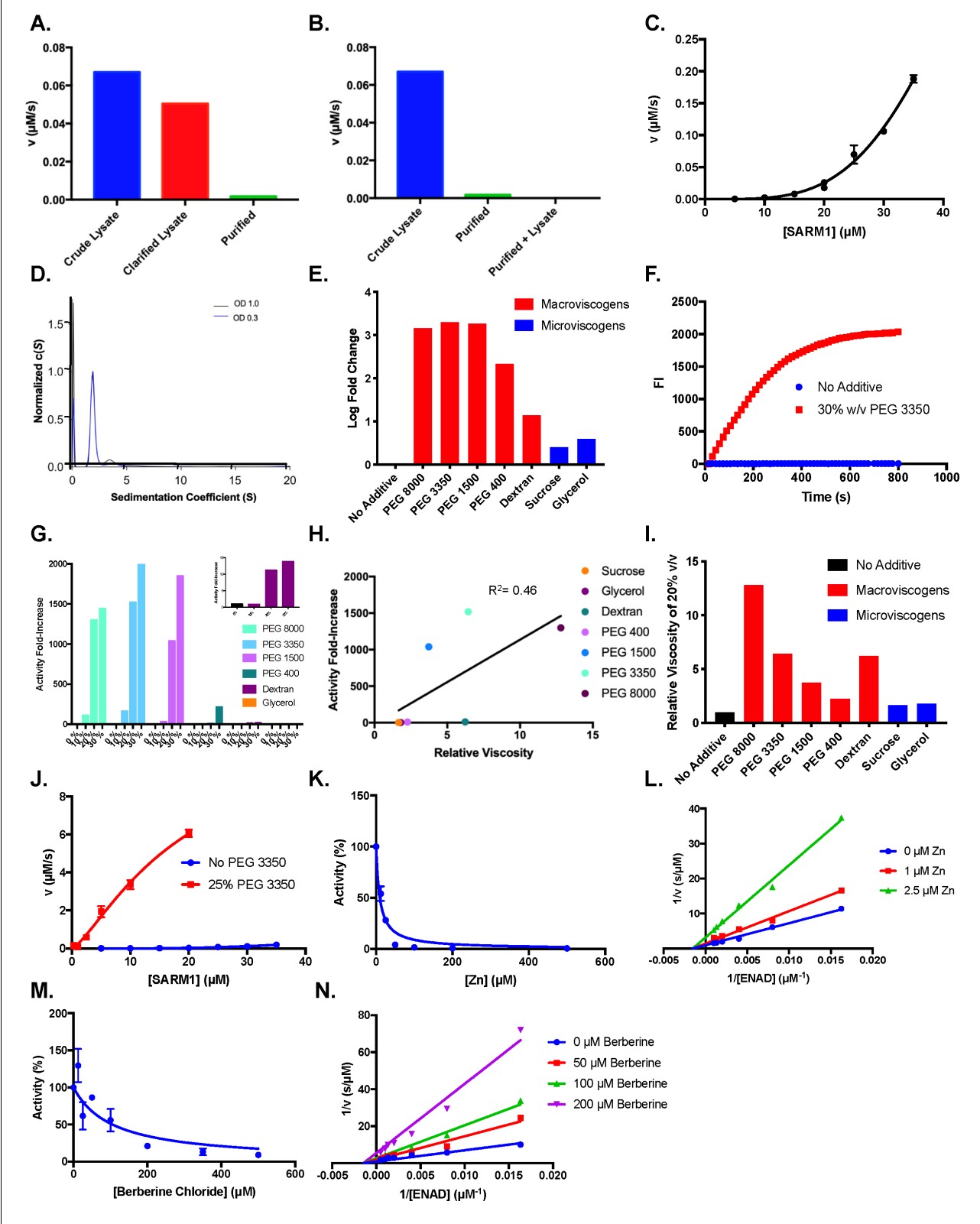

**Figure 2.** Effect of crowding agents on activity. (**A**) Activity of crude lysate, clarified lysate, and elution during SARM1 purification (n=2). (**B**) Activity of crude lysates, pure protein and pure protein added back to C43 lysates (n=2). (**C**) Dose dependence of SARM1 TIR domain activity from 5 to 35 µM with 1 mM ENAD (n=3). (**D**) AUC of SARM1 at an $OD_{280}$ of 0.3 and 1.0 (n=3). (**E**) Log fold change in SARM1 (10 µM) activity in buffer compared to 30% of crowding agent and 1 mM ENAD (n=3). (**F**) Representative time course of SARM1 TIR domain with and without 30% w/v PEG 3350 at 2 mM ENAD

*Figure 2 continued on next page*

Figure 2 continued

and 20 µM SARM1 (n=3). (G) Dose dependence of SARM1 TIR domain activity from 0 to 30% crowding agent and 1 mM ENAD (n=3). (H) Activity of SARM1 (10 µM) at different viscosities of 20% w/v solutions (n=3). (I) Relative viscosities at 20% w/v of different solutions (n=2). (J) Dose dependence of SARM1 TIR domain activity from 300 nM to 35 µM with and without PEG (n=3). (K) Zinc inhibition of SARM1 (2.5 µM) plus 25% PEG 3350 with 100 µM ENAD (n=3). (L) Mechanism of zinc inhibition of SARM1 (2.5 µM) in the presence of 25% PEG 3350 (n=3). (M) Berberine chloride inhibition of SARM1 (2.5 µM) plus 25% PEG 3350 with 100 µM ENAD (n=3). (N) Mechanism of berberine chloride inhibition of SARM1 (2.5 µM) in the presence of 25% PEG 3350 (n=3).

The online version of this article includes the following figure supplement(s) for figure 2:

**Figure supplement 1.** Effect of additives on viscosity and activity.

**Figure supplement 2.** Effect of divalent cations on activity.

enzyme needed to reach this threshold. One potential explanation is that PEG induces molecular crowding by excluding water from the protein surfaces, thereby promoting multimerization and catalysis. Along these lines, we note that in the absence of PEG the Hill coefficient is 3.8, indicating a high degree of cooperativity, whereas in the presence of PEG (0–25%) the Hill coefficient decreases to 1.3, suggesting that PEG decreases the dependence on the cooperative formation of an active multimer (*Figure 2—figure supplement 1C–D*).

Next, we evaluated the potency of several metals that were previously shown to inhibit SARM1 when assayed in lysates (*Loring et al., 2020a*). Notably, $NiCl_2$ and $ZnCl_2$ completely eliminated activity in the presence of PEG (*Figure 2—figure supplement 2A*). This result is consistent with our previous data (*Loring et al., 2020a*; *Loring et al., 2020b*). Expanding on these studies, we also evaluated several other metal ions including $CaCl_2$, $CdCl_2$, $CoCl_2$, $CuCl_2$, $MgCl_2$, and $MnCl_2$ (*Figure 2—figure supplement 2A*). Of note, $CdCl_2$, $CoCl_2$, and $CuCl_2$, decreased activity in the presence of the pure protein but not in lysates. This inconsistency could be due to elements in the lysate preferentially sequestering these metals and thereby preventing their ability to act on SARM1. Given the high potency of $CdCl_2$ and $CuCl_2$, we used these inhibitors to titrate the enzyme and determine the percentage of active enzyme (*Figure 2—figure supplement 2B*). Based on these data, all the enzyme appears to be active. Although speculative, the fact that thiophilic metals (i.e. $CdCl_2$, $CoCl_2$, $CuCl_2$, $NiCl_2$, and $ZnCl_2$) inhibit SARM1 suggests the presence of a catalytically important thiol. Along these lines, we previously showed that C629 and C635, which are present in the SARM1 specific loop, are critical for activity in the absence of PEG (*Loring et al., 2020b*).

We next determined an $IC_{50}$ value for $ZnCl_2$ as this compound was previously shown to potently inhibit the SARM1 TIR domain assayed in lysates. Consistent with our prior data, the $IC_{50}$ value obtained in the presence of PEG is 10 ± 1 µM, which is very similar to that in lysates (10 ± 3 µM) (*Loring et al., 2020a*), and with purified protein without PEG (10 ± 1 µM) (*Loring et al., 2020b*; *Figure 2K*, *Table 1*). To further confirm that the addition of PEG 3350 was not introducing any confounding variables, we repeated the mechanism of inhibition studies with $ZnCl_2$. As before, we found that $ZnCl_2$ inhibits SARM1 noncompetitively with a $K_i$ value of 1.0 ± 0.1 µM. This value is similar to the value obtained when lysate derived SARM1 was assayed ($K_i$ = 3.3 ± 0.1 µM *Figure 2L*; *Loring et al., 2020b*).

For a secondary confirmation that PEG does not impact the in vitro properties of SARM1, we tested a different inhibitor, berberine chloride that was previously identified from a high-throughput screen (*Loring et al., 2020b*). Berberine chloride inhibited SARM1 in the presence of PEG with an $IC_{50}$ value of 140 ± 30 µM. This value is quite similar to its potency in lysates (140 ± 20 µM) and when assayed with purified SARM1 without PEG 3350 (140 ± 20 µM) (*Figure 2M* and *Table 1*;

**Table 1.** Comparison of Inhibitor Potency.

Assays were performed in triplicate at SARM1 concentrations of 300 nM[a], 20 µM[b], and 2.5 µM[c].

| | Lysate $IC_{50}$ (µM)[a] | Pure protein $IC_{50}$ (µM)[b] | Pure protein + PEG $IC_{50}$ (µM)[c] |
|---|---|---|---|
| Zinc | 10 ± 3 | 10 ± 1 | 10 ± 1 |
| Berberine Chloride | 140 ± 20 | 140 ± 20 | 140 ± 30 |

*Loring et al., 2020b*). As before, we found that berberine chloride inhibits SARM1 noncompetitively in the presence of PEG 3350 with a $K_i$ value of 120 ± 10 µM. This value and inhibition pattern are identical to that obtained when SARM1 was assayed in lysates ($K_i$ = 120 ± 10 µM, *Figure 2N*; *Loring et al., 2020b*).

We next sought to investigate how PEG enhances the activity of SARM1. Since PEGs are typically used as precipitants, we hypothesized that PEG3350 was precipitating and activating SARM1. Consistent with this hypothesis, we found that the addition of PEG 3350 (25% w/v) results in the formation of an active precipitate, whereas the protein in buffer without PEG remains soluble (*Figure 3A*). These data indicate that the observed rate enhancement is driven by a phase transition. Next, we evaluated the effect of protein and PEG concentration on this phase transition. We found that both enzyme (5–20 µM) and PEG (0–25%) concentration enhanced precipitate formation (*Figure 3B–C*). As the concentration of SARM1 in the supernatant decreases, the precipitate concentration increases with a corresponding increase in enzyme activity. This effect is evident by the absence of a pellet for 0% PEG, a faint band at 10% PEG, and darker bands at 17.5% and 25% PEG at 5, 10, and 20 µM SARM1, and the inverse trend for the supernatant (*Figure 3C*). To determine whether other precipitants increase SARM1 activity, we screened several common precipitants and found that citrate both precipitated and acted as a second highly potent activator of SARM1 activity (*Figure 3—figure supplement 1A*).

To explore the nature of the interactions involved in precipitate formation, we tested the effect of 1,6–hexanediol, an aliphatic alcohol that is often used to distinguish between hydrophobic interactions involved in liquid-to-liquid phase separations and liquid-to-solid phase transitions (*Kroschwald et al., 2015*; *Patel et al., 2007*; *Peskett et al., 2018*). We found that while 1,6–hexanediol inhibited the activity of pure protein alone, it did not substantially affect the activity of SARM1 in the presence of PEG or sodium citrate (*Figure 3D–E*). These data suggests that the activity of the pure protein is dominated by weak hydrophobic interactions or a liquid-to-liquid phase separation (*Figure 3D*). On the other hand, the fact that 1,6-hexanediol does not eliminate the activity of pure protein in the presence of PEG or sodium citrate, supports SARM1 undergoing a liquid-to-solid phase transition in these cases (*Figure 3E*). To confirm that precipitate formation is reversible, we performed the centrifugation experiment in the presence of PEG or sodium citrate and tested the activity of the fractions (pre-centrifugation, supernatant, precipitate resuspended in buffer, and precipitate resuspended with buffer plus additive) (*Figure 3—figure supplement 1B*). The lack of both a precipitate and activity in the precipitate resuspended in buffer confirm that precipitation is reversible. To see if select additives/inhibitors affect precipitation of SARM1, we quantified the precipitation percentage with PEG or citrate in the presence of Zn, NMN, Ca, and berberine and found that none of these additives affected precipitation (*Figure 3—figure supplement 1C–D*).

Having established that both PEG 3350 and sodium citrate enhance SARM1-mediated $NAD^+$ hydrolysis via a phase transition, we next sought to investigate how this phase transition enhances activity. To that end, we performed negative stain EM in the absence and presence of sodium citrate (500 mM) to determine how the precipitant affects activity and oligomeric state (*Figure 3F–G*). Surprisingly, we were able to detect the SARM1 TIR domain even in the absence of additive, which suggests the TIR domain alone is forming a higher ordered oligomer. The median diameter of the particles is ~8 nm or ~80 Å, which closely aligns with the diameter predicted for an octameric ring of the TIR domain; recent crystal and cryoEM structures of the SAM domains and full-length protein, respectively, show that SARM1 forms an octameric ring (*Bratkowski et al., 2020*; *Sporny et al., 2020*; *Sporny et al., 2019*). In the presence of sodium citrate, the SARM1 TIR domain forms significantly larger ring–like structures, potentially by wrapping around on itself (*Figure 3G*, p<0.00001). Notably, the mean area is 172 $nm^2$ (SEM = 5.2, n=100) in the presence of citrate compared to 66 $nm^2$ (SEM = 1.9, n = 100) without additive (*Figure 3H*). These data suggest that precipitants like PEG 3350 and sodium citrate enhance SARM1 activity by inducing the formation of a higher ordered oligomer and facilitating inter–TIR domain contacts necessary for $NAD^+$ hydrolysis.

We next sought to exploit the additive–induced rate enhancement to perform the first detailed analysis of SARM1 catalysis. First, we determined how pH affects activity and precipitate formation. We found that the catalytic efficiency ($k_{cat}/K_m$) peaks from pH 6–8, which is primarily due to an effect on $k_{cat}$, as the $K_m$ value remains nearly constant across the range tested (*Figure 3I*). Furthermore, the effect of pH on activity appears to be independent of precipitation as the precipitate percentage varies by only 10% across the range tested. Notably, the pH profile for $k_{cat}$ is bell-shaped with $pK_a$

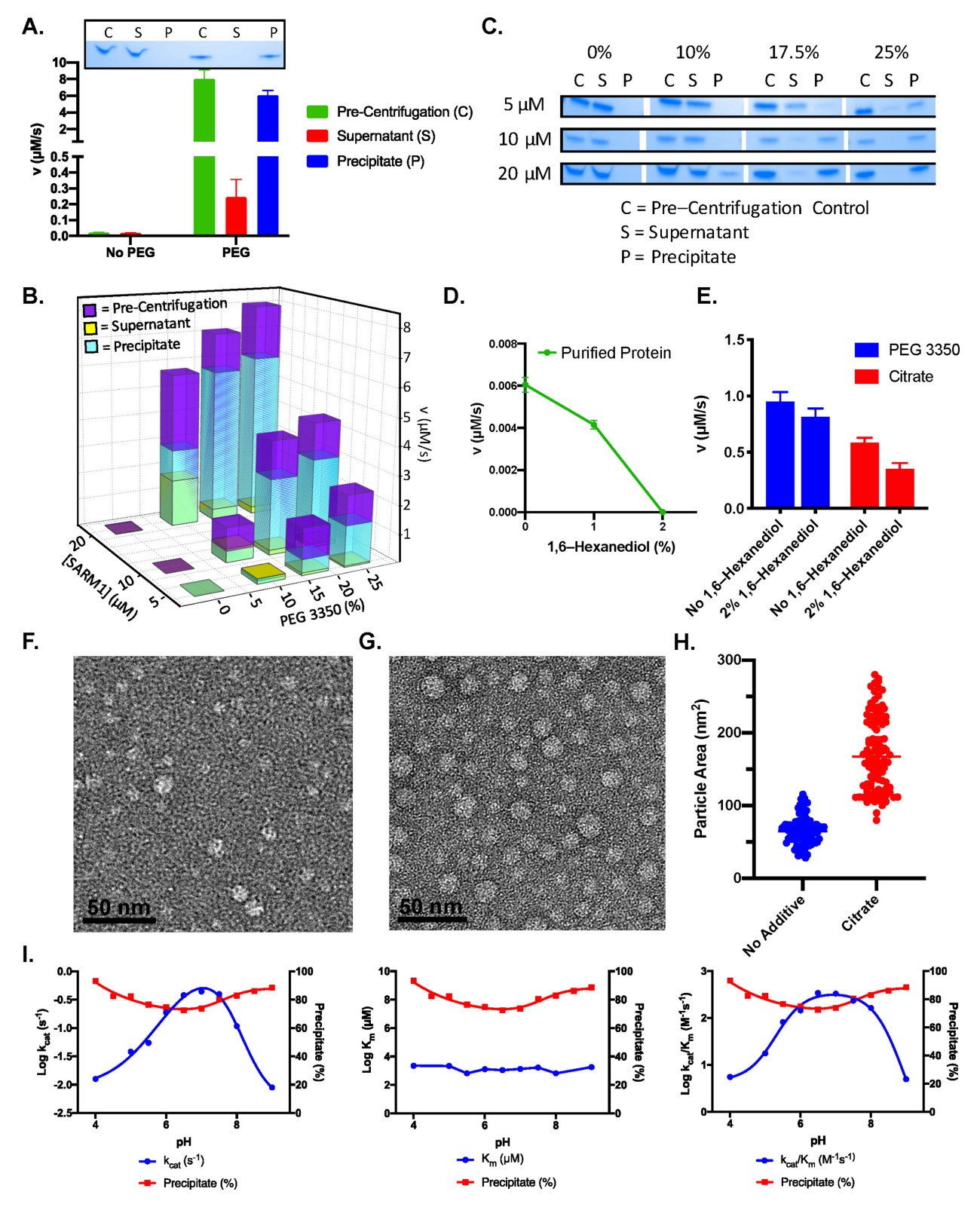

**Figure 3.** PEG 3350 induces a phase transition. (**A**) Activity of pre-centrifugation, supernatant, and precipitate fractions obtained before and after treatment of 20 µM SARM1 with and without 25% PEG 3350. Coomassie gels demonstrate the presence of SARM1 in each fraction (n=3). (**B**) Activity of pre-centrifugation, supernatant, and precipitate fractions obtained after treating SARM1 (5, 10, and 20 µM) with 0%, 10%, 17.5%, and 25% PEG 3350 (n=3). (**C**) Representative Coomassie-stained gels demonstrate the presence of SARM1 in each fraction analyzed in panel B (n=3). (**D**) Effect of 1,6–

*Figure 3 continued on next page*

Figure 3 continued

Hexanediol on pure SARM1 (20 µM) activity (n=3). (**E**) Effect of 2% 1,6–hexanediol treatment on SARM1 activity in the presence of 25% PEG or 500 mM sodium citrate (n=3). (**F**) Negative stain EM of SARM1 TIR domain in 50 mM HEPES pH 8.0, plus 150 mM NaCl. (**G**) Negative stain EM of SARM1 TIR domain in 50 mM HEPES pH 8.0, plus 150 mM NaCl and 500 mM sodium citrate. (**H**) Particle area in square nanometers quantified with ImageJ, mean is shown by line (No additive: Mean = 66.0, SEM = 1.9, n = 100; Citrate: Mean = 172.1, SEM = 5.2, n=100). Significance determined by T–test (*** = p value < 0.00001). (**I**) Effect of pH on the steady state kinetics and precipitate formation. Data is from an average of three experiments each conducted in duplicate (n=3). Full gels provided in *Figure 3—figure supplement 2*.

The online version of this article includes the following figure supplement(s) for figure 3:

**Figure supplement 1.** Additives precipitate and activate SARM1.

**Figure supplement 2.** Full gels of those shown in *Figure 3* and *Figure 3—figure supplement 1*.

values of $4.9 \pm 0.2$ and $8.8 \pm 0.2$ for the ascending and descending limbs, respectively. The $pK_a$ value for the ascending limb is consistent with the previous suggestion that a glutamate, likely E642, acts as a key catalytic residue (*Essuman et al., 2017*).

We next evaluated the effect of PEG on the steady state kinetic parameters in the presence of 10 µM SARM1. In the presence of 25% PEG, the $K_m$ decreases 3.3-fold, whereas $k_{cat}$ and $k_{cat}/K_m$ increase by 2000- and 7000-fold, respectively (*Figure 4A*). The fact that the increase in $k_{cat}/K_m$ is dominated by an increase in $k_{cat}$, suggests that PEG promotes multimerization and activity via precipitate formation. To further interrogate how PEG impacts the activity of SARM1, we performed detailed steady state kinetic experiments at various PEG concentrations from 0 to 25% (*Figure 4B*). We found that increasing concentrations of PEG 3350 cause the $K_m$ value to decrease and level off at ~500 µM (*Figure 4C*) and the $k_{cat}$ to increase almost linearly (*Figure 4D*). The catalytic efficiency also increases and plateaus at ~1700 $M^{-1}s^{-1}$ (*Figure 4E*, *Supplementary file 1a*). Interestingly, this value is nearly identical to that recorded in the lysate and on-bead (1500 $M^{-1}s^{-1}$) (*Essuman et al., 2017*; *Loring et al., 2020a*; *Figure 4B–E*, *Table 2*).

We also performed steady-state kinetic analyses at different concentrations of SARM1 in the absence and presence of 25% w/v PEG 3350 (*Figure 4F–J*). As with increasing PEG concentration, an increase in SARM1 concentration led to a small decrease in $K_m$ and an increase in $k_{cat}$ (*Figure 4F–J*, *Supplementary file 1b*), such that the $k_{cat}/K_m$ values are comparable to those obtained for the enzyme assayed in lysate (*Loring et al., 2020a*). Of particular note is the fact that PEG makes kinetic studies feasible at 500 nM of SARM1, which is near the concentration studied in lysates (300 nM) (*Loring et al., 2020a*). Specifically, at 500 nM of SARM1 with PEG, the $K_m$ is 1600 µM, the $k_{cat}$ is 0.24 $s^{-1}$, and the catalytic efficiency is 150 $M^{-1}s^{-1}$ (*Figure 4F–J*, *Supplementary file 1b*). Notably, the $K_m$ value at 500 nM with PEG is similar to that at 20 µM SARM1 without PEG (1200 µM); however, the $k_{cat}$ is ~100-fold greater in the presence of PEG (*Figure 4F–J*, *Supplementary file 1b*). These data suggest that by inducing a phase transition, PEG drastically enhances enzymatic turnover, but has little effect on substrate binding. Although the $K_m$ value slightly decreases as SARM1 or PEG concentration increases, it remains much higher than the lysate value of $40 \pm 10$ µM (*Loring et al., 2020a*), indicating that PEG/SARM1 concentration does not completely recapitulate SARM1 activity in the lysate or on-bead (*Figure 4*, *Table 2*). The similarities in $k_{cat}/K_m$, however, suggest compensation (i.e. alterations in rate limiting steps) between these two parameters. Moreover, the fact that $k_{cat}$ is directly correlated with enzyme and PEG concentration, suggests that turnover rate is largely dependent on the multimeric state and the formation of an ordered oligomer. While the $k_{cat}/K_m$ values obtained for SARM1 (~2000 M-1s-1) are not particularly robust, they are in-line with other eukaryotic enzymes that we have studied (e.g. the protein arginine deiminases, protein arginine methyltransferases, and nicotinamide N-methyltransferase *Knuckley et al., 2007*; *Knuckley et al., 2010*; *Loring and Thompson, 2018*; *Osborne et al., 2007*), and the slow kinetics likely relate to the fact that degeneration occurs on the hour time scale (*Mack et al., 2001*; *Osterloh et al., 2012*).

Next, we evaluated the product specificity of SARM1 with PEG and sodium citrate using an orthogonal LC-MS-based assay. We found that the addition of PEG or citrate led to a time-dependent decrease in $NAD^+$ levels that coincided with increased levels of nicotinamide and a mixture of ADPR and cADPR (*Figure 5A–B*, *Figure 5—figure supplement 1A–B*). Of note, the ADPR to cADPR ratios are consistent over time under each condition (*Figure 5C*), indicating that cADPR is a poor substrate relative to $NAD^+$. Interestingly, the addition of PEG increases the production of ADPR

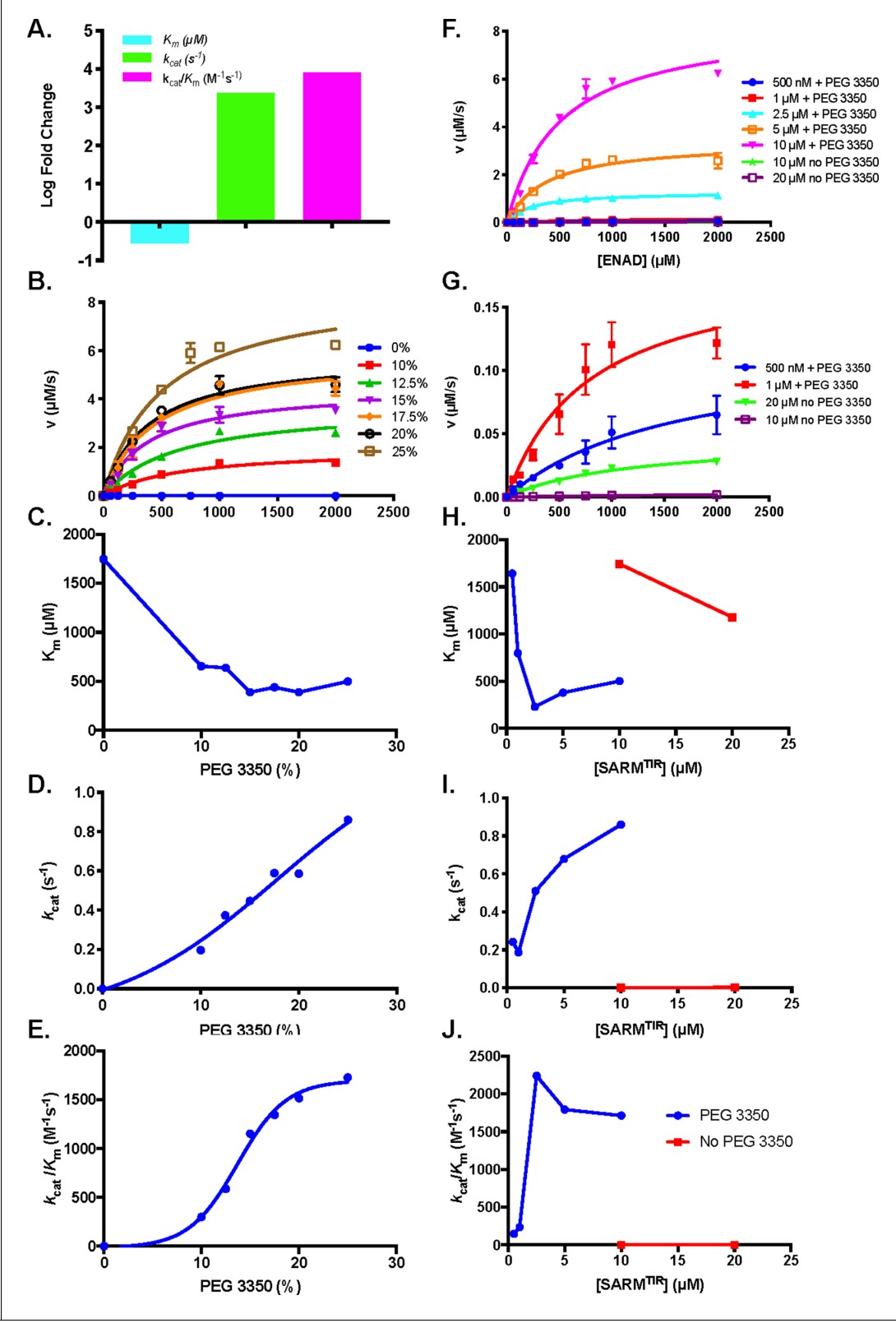

**Figure 4.** Effect of PEG 3350 concentration on steady state kinetics of ENAD hydrolysis. (**A**) Steady state kinetic analysis at 10 µM SARM1 with and without addition of PEG 3350 (25%) (n=3). (**B**) Steady state kinetic analysis of SARM1 (10 µM) with PEG 3350 (0–25% w/v) (**C**) $K_m$, (**D**) $k_{cat}$, and (**E**) $k_{cat}/K_m$ values at each concentration (n=3). (**F**) Steady state kinetic analysis of SARM1 from 500 nM to 10 µM with PEG 3350, and without PEG 3350 at 10 and 20 µM from 0 to 2 mM ENAD, fit to Michaelis Menten equation (n=3). (**G**) Blow up of steady state kinetic analysis of SARM1 500 nM and 1 µM with PEG

*Figure 4 continued on next page*

Figure 4 continued

3350, and 10 and 20 µM without PEG 3350 with ENAD from 0 to 2 mM. Fitting to Michaelis Menten equation gives (H) $K_m$, (I) $k_{cat}$, and (J) $k_{cat}/K_m$ values at each concentration (n=3).

relative to cADPR from a ratio of $10 \pm 2$ to a ratio of $16 \pm 1$. This ratio is further increased in the presence of citrate to $23 \pm 3$ (*Figure 5D*). We hypothesize that this could be due to the multimeric state of the enzyme constraining $NAD^+$ in such a position that it favors hydrolysis by water as compared to reaction with the N1 of the adenosine, which is required for cyclization. We also evaluated whether SARM1 could cleave cADPR in the presence or absence of additive (*Figure 5—figure supplement 2A–C*). Consistent with our prior finding that SARM1 follows an ordered uni–bi kinetic mechanism instead of a sequential intermediate mechanism, SARM1 is unable to cleave cADPR in the absence of additive and in the presence of sodium citrate (*Loring et al., 2020a*; *Figure 5—figure supplement 2A–C*). Further support comes from a single turnover experiment where SARM1 concentration exceeded the $NAD^+$ concentration and the ADPR/cADPR ratio remains nearly constant over the time range observed, indicating that SARM1 does not cleave cADPR under these conditions (*Figure 5—figure supplement 3A–B*). Interestingly, however, SARM1 slowly cleaves cADPR in the presence of PEG, resulting in the time-dependent production of ADPR (*Figure 5—figure supplement 3C–D*). Additionally, we found that cADPR inhibits SARM1 in the presence of PEG with an $IC_{50}$ of $100 \pm 10$ µM and a competitive inhibition pattern, supporting its role as a poor substrate under these conditions (*Figure 5—figure supplement 3E–F*).

To further interrogate the substrate specificity of SARM1, we analyzed the capacity for SARM1 to cleave 16 $NAD^+$ analogues (*Figure 5—figure supplement 4*). Notably, we found that adenosine substitutions are well tolerated as etheno–NAD (ENAD), nicotinamide guanine dinucleotide (NGD) and nicotinamide hypoxanthine dinucleotide (NHD) are all processed by SARM1. By contrast, SARM1 does not cleave NADH or NHDH, the reduced forms of $NAD^+$ and NHD, in the absence or presence of PEG. This finding likely relates to nicotinamide being a better leaving group than its reduced form, and/or conformational preferences; nicotinamide is planar whereas the reduced form adopts a puckered conformation. Along these lines, SARM1 can still cleave nicotinamide adenine dinucleotide phosphate ($NADP^+$) but cannot accommodate the flavin substitution for nicotinamide in flavin adenine dinucleotide (FAD). SARM1 can also cleave thionicotinamide adenine dinucleotide (sNAD), which has a thiol substitution for the carbonyl. Interestingly, low level cleavage was observed for nicotinamide mononucleotide (NMN), but not for nicotinic acid mononucleotide (NaMN) nor nicotinic acid adenine dinucleotide (NaAD), suggesting that SARM1 cannot accommodate a carboxylic acid substitution at this position. However, SARM1 can accommodate a methyl substitution at the same site as it will cleave 3-acetylpyridine adenine dinucleotide (3apAD). Adenosine triphosphate (ATP), adenosine diphosphate (ADP), guanosine triphosphate (GTP) and ara-F-$NAD^+$ were also not cleaved by SARM1, nor did they inhibit SARM1 cleavage of $NAD^+$. Ara-F-$NAD^+$ has a fluorine substitution for a hydroxyl with opposite stereochemistry, which likely interferes with coordination to the active site as no cleavage or inhibition was evident.

Next, we generated mutant constructs to understand how specific residues contribute to catalysis and multimerization. Residues were selected based on their proposed roles in multimerization, catalysis, or location in the SARM1 specific loop (*Figure 5E*). All the mutants examined showed a marked reduction in catalytic efficiency, $k_{cat}/K_m$, ranging from 100- to 2000-fold (*Figure 5F*). Interestingly, the most extreme effect was observed for the E642A mutant, which has previously been suggested

**Table 2.** Comparison of steady-state kinetic parameters for the SARM1 TIR domain.

| | 2.5 µM PEG | Lysate[*] | On-Bead[*] | Literature[†] |
|---|---|---|---|---|
| $K_m$ (µM) | $230 \pm 30$ | $40 \pm 10$ | $28 \pm 6$ | 24 |
| $k_{cat}$ ($s^{-1}$) | $0.51 \pm 0.01$ | $0.06 \pm 0.03$ | $0.04 \pm 0.03$ | 0.17 |
| $k_{cat}/K_m$ ($M^{-1}s^{-1}$) | $2200 \pm 30$ | $1500 \pm 300$ | 1500 | 7083 |

[*](*Loring et al., 2020a*).
[†]Reported by *Essuman et al., 2017* with SARM1 on-bead by LC-MS.

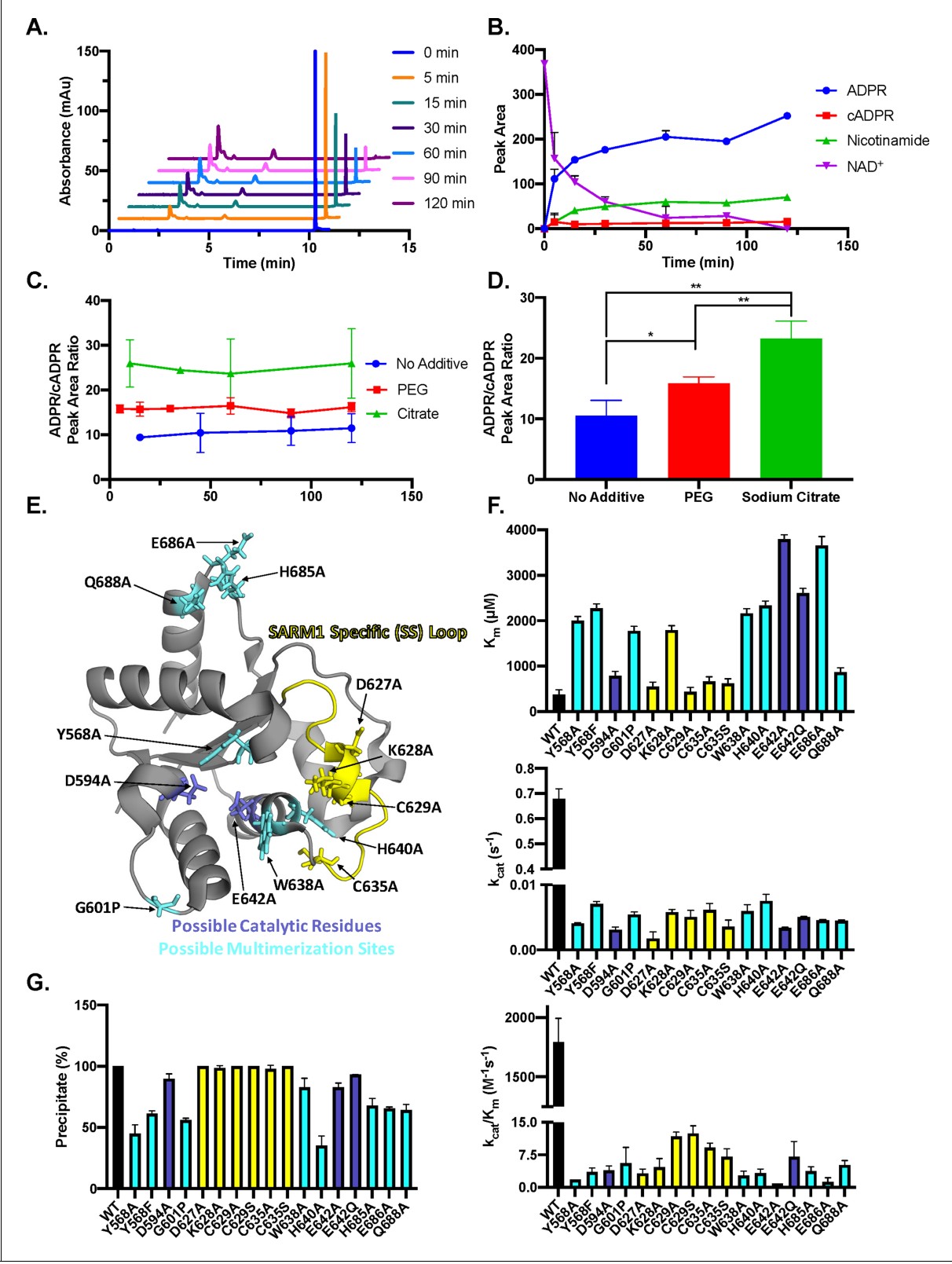

**Figure 5.** SARM1 reaction products and mutagenesis. (A) Representative time course of SARM1 with PEG, depicting the $A_{254}$ nm for ADPR, cADPR, nicotinamide, and $NAD^+$ (n=3). (B) Peak area quantification of $NAD^+$ and reaction products ($A_{254}$ nm) over time (n=3). (C) ADPR/cADPR peak area ratios at $A_{254}$ over time without additive, with PEG or with sodium citrate (n=3). (D) Average of ADPR/cADPR peak area ratios shown in panel C for no additive, PEG and sodium citrate SARM1 reactions, where * = p < 0.0005 and ** = p < 0.00005. (E) Mutants of interest shown on the SARM1 TIR

*Figure 5 continued on next page*

*Figure 5 continued*

domain structure, colored yellow for SARM1-Specific loop, dark blue for potential catalytic residues, and light blue for potential multimerization sites (PDBID: 6o0q). (**F**) $K_m$, $k_{cat}$, and $k_{cat}/K_m$ values obtained from steady-state kinetic experiments performed in duplicate for the mutants compared to wild-type SARM1 TIR domain, colored yellow for SARM1-Specific loop, dark blue for potential catalytic residues, and light blue for possible multimerization sites (n=2). (**G**) Precipitation of mutants compared to wild-type SARM1 TIR domain performed in duplicate (n=2).

The online version of this article includes the following figure supplement(s) for figure 5:

**Figure supplement 1.** SARM1 LC–MS reactions.
**Figure supplement 2.** Hydrolysis of cADPR.
**Figure supplement 3.** Hydrolysis of cADPR.
**Figure supplement 4.** Evaluation of potential substrates with and without PEG 3350.

to be a key catalytic residue based on structural alignments with other glycohydrolases (*Essuman et al., 2017*; *Figure 5F*). Consistent with a role in multimerization the Y568, G601, H640, and H685 mutants showed reduced precipitation, suggesting that the loss in activity observed upon their mutation is due in part to a reduction in multimer formation (*Figure 5G*).

While $k_{cat}$ values were reduced by ~100- to 400-fold for all mutants examined, a subset showed a >4–fold increase in $K_m$, including residues thought to be involved in multimerization, that is, Y568, G601, W638, H640, and E686, and possible catalytic residues, K628 and E642. Several of these mutants have been previously characterized in the context of the purified protein and degenerating axons (*Horsefield et al., 2019*; *Summers et al., 2016*). Consistent with our results, they also found that TIR domain mutants E642A, Y568A, H685A, and BB loop mutations (D594A and G601P) reduce NAD$^+$ hydrolase activity (*Horsefield et al., 2019*). Moreover, several of these mutations, including G601P, D627K, K628D, and C629S, were sufficient to prevent axonal degeneration post–axotomy, indicating that they render SARM1 non-functional (*Summers et al., 2016*).

To investigate the multimeric state of SARM1 in cells, we transfected HEK293T cells with a GFP-tagged SAMTIR–E642A construct; this mutant construct should multimerize like wild-type SARM1 without exhibiting toxicity due to the rapid depletion of NAD$^+$ (*Figure 6A–B*). We found that this GFP tagged SARM1 SAMTIR construct forms puncta in cells (*Figure 6A–B*). To demonstrate that these puncta are due to SARM1 oligomerization we transfected cells with a construct encoding GFP–tagged SAMTIR-E642A plus an additional G601P mutation. G601 has been previously implicated as an association interface for TIR domains during SARM1 activation, and we demonstrated that the BB loop mutation G601P, reduces both activity and precipitation in vitro (*Figure 5F–G*). Moreover, the G601P mutation has previously been shown to block axon degeneration in cells (*Gerdts et al., 2013*; *Sporny et al., 2019*). Notably, the G601P point mutation also reduces SARM1 puncta formation in HEK293T cells (*Figure 6A–C*). Based on spot counts, the IDEAS software spot count wizard identified buckets of distinct sub–populations (0–28; R1 and R2). Specifically, cells expressing the E642A mutant show a higher frequency of puncta (34.4% of cells) whereas the majority of cells expressing the E642A/G601P double mutant were classified into the R1 or baseline fluorescence group (96.3%), with a smaller percentage exhibiting puncta characterized as the R2 group (3.7%) (*Figure 6A–C*). Since the SAM domains are involved in octamerization, which is not impacted by the G601P mutation, these data indicate that puncta formation is largely driven by the TIR domain.

Having determined that SARM1 oligomerizes in cells, we next sought to determine whether SARM1 oligomerizes in vivo and if so, investigate how oligomerization affects SARM1-mediated degeneration. To this end, we opted to study the SARM1 orthologue, TIR-1 in the GABA motor nervous system of *C. elegans*. *C. elegans* encodes several TIR-1 splice variants including TIR-1b and TIR-1d which lack the ARM domain (*Figure 6—figure supplement 1A*) but contain the tandem SAM domains essential for octamer formation. Since TIR-1b was used in our in vivo studies (see below), we expressed and purified an MBP-tagged ceSAMTIR expression construct as a mimic of TIR-1b and evaluated whether citrate activates this construct. As a control, we also evaluated the effect of citrate on the *C. elegans* TIR domain (ceTIR). Like the human TIR domain, the NAD$^+$ hydrolase activity of both ceTIR and ceSAMTIR was greater with citrate, where ceTIR and ceSAMTIR activity were enhanced 22-fold and 1.4-fold respectively (*Figure 6—figure supplement 1B*). While citrate does not activate the ceSAMTIR construct to the same extent as the ceTIR domain alone, mutations that

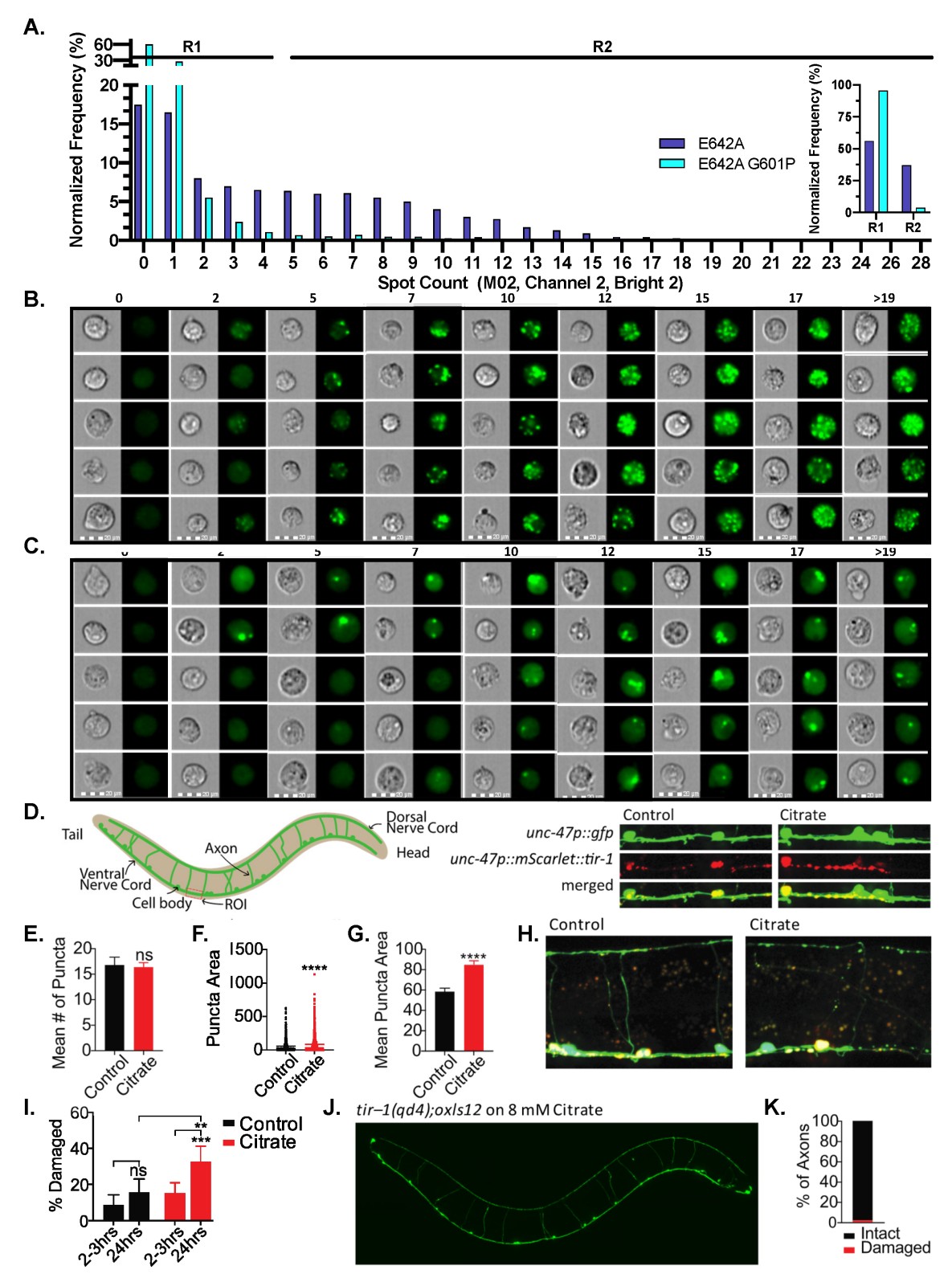

**Figure 6.** G601P disrupts SARM1 puncta formation in HEK293T cells and citrate induces the formation of larger puncta in *C.elegans*. (**A**) Normalized frequency of GFP-positive 7AAD-negative HEK293T cells expressing either GFP–tagged E642A (20,927 cells) or GFP–tagged E642A G601P (9895 cells) in the context of a SAMTIR construct. Images are bucketed according to spot count algorithm (IDEAS software), where R1 is designated not significantly different than baseline and R2 significant. Insert shows the percentage of cells for each mutant designated as R1 versus R2. (**B**)

*Figure 6 continued on next page*

*Figure 6 continued*

Representative images from each bucket for SAMTIR–E642A. (C) Representative images from each bucket for SAMTIR–E642A/G601P. (D) Diagram of *C. elegans* GABA motor nervous system and quantified region of interest (ROI). Representative micrographs of the ventral nerve cords of untreated control and 8 mM citrate-treated animals 2–3 hr post-citrate application. GABA motor neurons express cytoplasmic GFP and the mCherry tagged TIR-1 transgene (*unc-47p::GFP* and *unc-47p::mScarlet::tir-1b::let-858* 3'UTR, respectively). (E) Mean number of puncta in the ventral nerve cord ROIs of untreated animals placed on control plates and animals placed on citrate plates after 2–3 hr. The mean number of puncta is shown with SEM and significance determined by t-test (n=12 animals citrate and n=10 animals control). (F) Puncta area in citrate-treated worms compared to control animals after 2–3 hr. Each individual puncta is plotted along with mean, significance determined by t-test (n=902 citrate and n=739 control). (G) Mean puncta area in citrate treated compared to control animals after 2–3 hr. Mean puncta area is shown as mean and SEM, significance determined by t-test (n=902 citrate and n=739 control). (H) Representative micrographs of control and citrate-treated animals 2–3 hr post citrate application. Green is *unc-47p::GFP* neuronal reporter, red is *unc-47p::mScarlet::tir-1b::let-858* 3'UTR transgene. (I) Percentage of damaged axons in citrate treated animals compared to untreated controls at 2–3 hr (n=169 axons citrate and n=145 axons control) and 24 hr (n=111 axons citrate and n=115 axons control). Shown are 95% CI and significance determined by Fishers exact test. (J) Representative micrograph of citrate-treated *tir-1(qd4)* mutant animals 24 hr post citrate application. GABA neurons are visualized with *unc-47p::GFP*. (K) The percentage of intact and damaged axons in *tir–1(qd4)* mutant animals 24 hr post citrate application (n=160 axons).

The online version of this article includes the following figure supplement(s) for figure 6:

**Figure supplement 1.** ceSAMTIR undergoes a phase transition that increases activity.
**Figure supplement 2.** Flowsight of cells after FACS selection with GFP and 7AAD.

disrupt the TIR–ARM interface or deletion of the ARM domain activate SARM1 to a similar extent (i.e. 1.8- to 2.4-fold) (*Jiang et al., 2020*).

Having demonstrated that ceSAMTIR activity is enhanced by citrate, we sought to determine whether ceSAMTIR undergoes a phase transition. To do this, we incubated ceSAMTIR with increasing concentrations of citrate, centrifuged the samples, removed the soluble fraction, and resuspended the insoluble pelleted fraction. All fractions were analyzed by SDS-PAGE for the presence of ceSAMTIR. At citrate concentrations 250 mM and below, ceSAMTIR was predominantly located in the supernatant. By contrast, at citrate concentrations of 500 mM and above, the protein was increasingly found in the pellet, such that at 1000 mM citrate, ceSAMTIR was primarily located in the pellet. Moreover, enzymatic activity in the fractions correlated with the location of ceSAMTIR (*Figure 6—figure supplement 1C*). These data are consistent with the hypothesis that longer versions of SARM1/TIR-1 also undergo a phase transition that correlates with enzyme activation.

The *C. elegans* GABA motor nervous system is composed of neurons that have cell bodies along the ventral nerve cord and extend axon commissures circumferentially around the body of the worm to the dorsal nerve cord. We visualized the GABA motor nervous system and TIR-1 using a transgenic strain that expresses cytosolic GFP and mScarlet::TIR-1b specifically in GABA motor neurons (*Figure 6D*; *Julian and Byrne, 2020*). Consistent with our cell culture studies, TIR-1 forms puncta in axons under basal conditions (*Figure 6D*). Given our demonstration that citrate activates SARM1 in vitro, we sought to determine whether citrate could alter TIR-1 expression and puncta formation in vivo. For these studies, L4 stage worms were exposed to 8 mM citrate and imaged 2–3 hr and 24 hr later. After 2–3 hr, no difference was observed in the mean number of puncta between untreated control worms and worms that were exposed to citrate (*Figure 6E*). However, there was a significant increase in mean puncta area seen in citrate treated worms (mean=84.83, SEM=4.03, n=902) compared to untreated control animals (mean=58.42, SEM=3.45, n=739) (*Figure 6F–G*). Note that the increase in puncta size was not a secondary consequence of gross axon morphology as the presence and size of mScarlet puncta does not coincide with the expression pattern of cytosolic GFP (*Figure 6H*). Together these data support that citrate treatment not only enhances the activity of SARM1 by inducing oligomerization in vitro but also facilitates multimerization in vivo as evidenced by the formation of larger puncta in *C. elegans* axons.

Given that citrate induces the formation of larger TIR-1 puncta in *C. elegans* axons, we determined whether this translates into downstream axonal degeneration. To this end, axons were analyzed at 2–3 hr and again at 24 hr post-citrate treatment. Axon commissures that were broken or fragmented, truncated, or missing were counted as damaged (*Figure 6I*). No significant change in the percentage of damaged axons was observed in the citrate treated animals 2–3 hr post-treatment (*Figure 6I*). However, when the same animals were analyzed 24 hr post-treatment, there was a significant increase in the percentage of damaged axons in citrate treated animals compared to untreated

controls (*Figure 6I*). Note the fact that axons are significantly fragmented or degraded at 24 hr precludes quantification of puncta number and area at this time point. To confirm that citrate-induced damage was dependent on TIR-1 function, we exposed *tir–1(qd4)* mutant animals, which have a null mutation in *tir–1*, to citrate for 24 hr and quantified axon damage (*Figure 6J*). We found that 98% of axons remain intact while only 2% were damaged, indicating that citrate exposure induces TIR-1-dependent axonal damage (*Figure 6K*). In total, these findings indicate that citrate induces TIR-1 multimerization and axonal degeneration in *C. elegans*.

## Conclusions

Our data demonstrate that SARM1 activity is largely dependent on concentration and crowding alone and that as SARM1 multimerizes it becomes more active. Moreover, we show that SARM1 activity is enhanced by a phase transition in vitro. These data are consistent with cellular and in vivo data showing that SARM1 forms puncta and that puncta size increases in response to neuronal injury. These findings are notable because SARM1 was previously thought to function as a dimer. The formation of a higher ordered multimer is, however, consistent with recent structural studies showing that the SAM domains in isolation and in the context of the full-length enzyme form octamers (*Bratkowski et al., 2020*; *Gerdts et al., 2013*; *Horsefield et al., 2019*; *Jiang et al., 2020*; *Sporny et al., 2020*; *Summers et al., 2016*). However, the resolution of these structures was not sufficient to determine the active conformation of the TIR domains (*Bratkowski et al., 2020*), and it is unclear whether the TIR domains form dimers, tetramers, octamers, or larger multimers via two or more octameric rings interacting with one another (*Bratkowski et al., 2020*; *Sporny et al., 2020*). Our data suggests that TIR domain multimerization is required to upregulate SARM1 activity.

Significant cellular and biochemical data supports the notion that the ARM domain interacts with and inhibits the TIR domain. However, mutations that disrupt this interaction cause subtle increases (~2-fold) in SARM1 activity (*Bratkowski et al., 2020*; *Jiang et al., 2020*; *Sporny et al., 2020*). By contrast, our data demonstrates that SARM1 multimerization increases catalytic efficiency and turnover rate by >2000-fold. Pre-organization of the TIR domains within the context of the octameric structure likely increases the effective concentration of the TIR domain such that multimerization can occur more readily. This conclusion is supported by our finding that our MBP-tagged ceSAMTIR expression construct has a higher basal level of activity that can be further increased by the addition of citrate. Moreover, the data suggest that the ARM domain prevents TIR domain multimerization and limits full activation of the enzyme.

Our observation that SARM1 becomes progressively more active as it undergoes a phase transition and forms higher ordered multimers, is analogous to other systems including purine biosynthesis where higher ordered oligomers, known as purinosomes, form under certain conditions to enhance purine synthesis (*An et al., 2010*; *Chan et al., 2015*). Moreover, SARM1 multimerization is reminiscent of the structures that are formed by inflammasomes and apoptosomes during an immune response or apoptosis. Interestingly, other TIR domain–containing proteins (e.g. MAL) form insoluble fibrils (*Ve et al., 2017*).

Supporting the notion that SARM1 multimerizes, we also show that SARM1 forms puncta in cells and that mutations in the BB loop interface (i.e. G601P) is sufficient to disrupt puncta formation. Notably, punctate expression of SARM1 in neurons (both soma and dendrites) has been observed in numerous studies (*Chen et al., 2011*; *Osterloh et al., 2012*), further confirming that SARM1 can form higher ordered structures. In the context of a degenerating axon, TIR domain multimerization, could translate into a feed-forward mechanism to induce degeneration. In fact, we demonstrate that increasing TIR–1/SARM1 multimerization in *C. elegans* via citrate treatment is sufficient to induce axonal degeneration. Furthermore, SARM1 protein levels have been documented to increase significantly in response to injury prior to Wallerian degeneration (*Massoll et al., 2013*), which is consistent with our data demonstrating that increasing SARM1 concentration enhances activity and inducing multimerization in vivo causes axonal degeneration. These data are also consistent with a prior report showing that upregulation of SARM1 is sufficient to induce axonal degeneration (*Gerdts et al., 2013*).

Based on these studies, we propose a model whereby, increasing SARM1 protein levels induces TIR domain multimerization that enhances SARM1 activity via a phase transition and that this process is mimicked by PEG 3350 and sodium citrate (*Figure 7*). We hypothesize that the enhanced activity associated with this phase transition is due to the formation of a higher ordered oligomer as we see

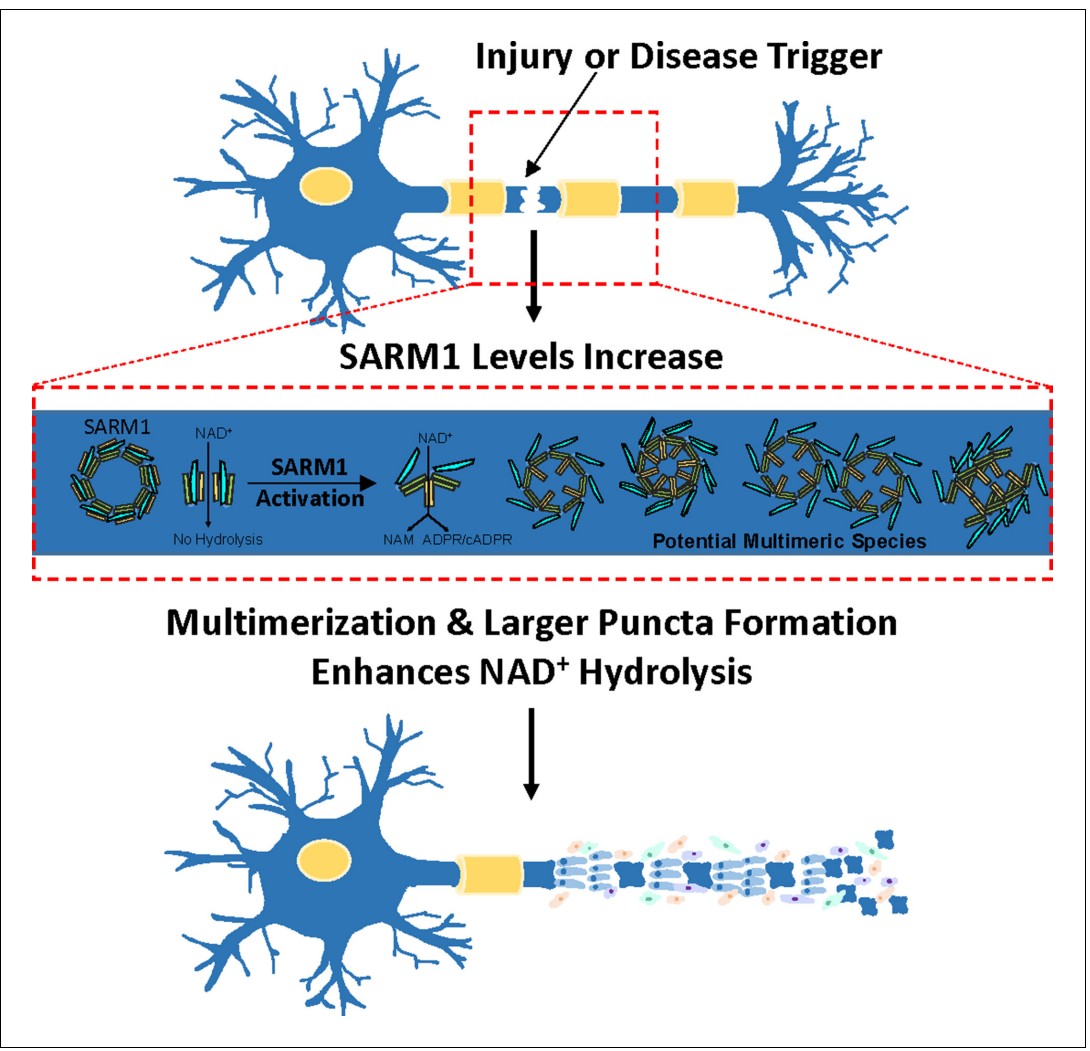

**Figure 7.** Model for precipitant-mediated enhancement in SARM1 activity and subsequent axonal fragmentation.

larger multimers by negative stain, formation of puncta in HEK293T cells, and induction of larger puncta with citrate treatment in *C. elegans*. We expect that the formation of larger puncta facilitates increased inter-TIR domain contacts that allow for enhanced NAD$^+$ hydrolysis and subsequent axonal degeneration. Similar phase transitions have been observed in multiple signaling pathways. For example, the liquid phase condensation of cyclic GMP–AMP synthase (cGAS) activates innate immune signaling (*Du and Chen, 2018*) and liquid-to-solid phase transitions have been reported for FUS and huntingtin, during the formation of aggregates associated with ALS and Huntington's disease (*Patel et al., 2015*; *Peskett et al., 2018*). Given that citrate is a mitochondrial metabolite, it is tempting to speculate that citrate, either from stressed mitochondria or from inhibition of the cytosolic enzyme ATP-citrate lyase, might promote a similar phase transition in vivo. However, the concentrations of citrate in plasma and cerebrospinal fluid are on the order of 0.1–0.4 mM (*Costello and Franklin, 2013*; *Westergaard et al., 2017*) versus the mM levels of citrate required to activate SARM1 in vitro. Perhaps in the context of a living cell a lower threshold of citrate could promote this phase transition. Consistent with the latter possibility is our demonstration that lower amounts of citrate promote puncta formation and degeneration in vivo. Further research is needed to fully address this possibility.

In summary, we show that SARM1 catalytic activity is enhanced by TIR domain multimerization and a phase transition reminiscent of that documented with other neurodegenerative disease-causing proteins. Therefore, inhibiting multimerization or this phase transition could be a target for

future therapeutic development for SARM1-associated diseases. In the context of a degenerating axon, we rationalize this phase transition as SARM1 exhibiting gradual NAD$^+$ hydrolase activity upon initial activation (which is demonstrated by in vitro ARM domain mutations relieving inter-TIR ARM domain contacts) (*Bratkowski et al., 2020*; *Jiang et al., 2020*) until a critical threshold is reached whereby SARM1 levels increase allowing SARM1 to multimerize and form larger puncta. We anticipate that higher ordered multimerization translates into increased activity and rapid depletion of NAD$^+$, which further relieves autoinhibition by NAD$^+$ (*Jiang et al., 2020*) and causes the axon to undergo catastrophic fragmentation and granular disintegration. This could shed light on a built–in protective mechanism to control the toxic effects of SARM1; thus, ensuring the neuron has ample time to circumvent the degenerative path, but when committed to it, degeneration is executed in a positive feed-forward mechanism resulting in abrupt fragmentation and granular disintegration during the latter stages of Wallerian degeneration.

## Experimental section

### Recombinant protein expression and purification

A recombinant human SARM1 TIR domain construct was bacterially expressed and purified as previously described (*Loring et al., 2020a*; *Loring et al., 2020b*). Briefly, the TIR domain construct in the pET30a+ expression vector was transformed into chemically competent *Escherichia coli* C43 (DE3) cells and stored as a glycerol stock. Starter cultures were prepared by diluting glycerol stocks 1:1000 in LB media and grown at 37°C overnight. The next day these starter cultures were diluted 400–fold in LB containing 50 µg/mL kanamycin and grown at 37°C while shaking at 225 rpm until the culture reached an OD$_{600}$ of 0.8. Protein expression was then induced by the addition IPTG (0.5 mM) and the incubation temperature was reduced to 16°C for 16–18 hr. Cells were then harvested by centrifugation at 3000 x g for 15 min at 4°C. The supernatant was removed, and the pellet was resuspended in lysis buffer (100 mM HEPES pH 8.0, 200 mM NaCl, 10% glycerol, 0.01% Tween 20) with Pierce EDTA–free protease inhibitor tablets (Thermo Scientific). The lysate was sonicated at an amplitude of 6 for 30 s (oscillating for 1 s on and 1 s off) for a series of 12 cycles using a Fisher Scientific Sonic Dismembrator sonicator (FB–705). After sonication, the lysate was clarified by centrifugation at 15,000 x g for 30 min at 4°C. The clarified lysate was applied to pre-equilibrated Streptacin XT resin. The resin was then washed with Wash Buffer (50 mM HEPES pH 8.0 and 500 mM NaCl). SARM1 was eluted with Wash Buffer containing 50 mM biotin. The eluent was concentrated and injected on a HiLoad Superdec SUP75 (or SUP200) column for size exclusion chromatography using 50 mM HEPES pH 8.0 and 150 mM NaCl as the running buffer. Protein was concentrated using an Amicon 10 kDa cutoff centrifugal filter and protein concentration was determined using the Bradford assay.

The recombinant *C. elegans* TIR-1 TIR domain was expressed in bacteria as previously described (*Loring et al., 2020a*). Briefly, the TIR domain in a pET-30a(+) vector was transformed into chemically competent *E. coli* BL21(DE3) cells and maintained as a glycerol stock at −80°C. An inoculation loop was used to transfer the transformed bacteria into 5 mL of LB media with 50 µg/mL (final concentration) of kanamycin and the culture was grown overnight at 37°C while rotating. The next day, the cultures were diluted 1:400 in LB media with 50 µg/mL (final concentration) of kanamycin and grown at 37°C while shaking at 215 rpm until an OD$_{600}$ of 0.7–0.8 was reached. After cooling, 50 µM IPTG (final concentration) was added to the culture to induce protein expression. The incubator temperature was decreased to 16°C and cells were incubated for an additional 16–18 hr. Bacterial cells were collected by centrifugation at 3000x*g* for 15 min at 4°C, flash frozen in liquid nitrogen, and stored at −80°C until purification.

For purification, bacterial pellets were thawed on ice and then resuspended in Lysis Buffer (50 mM Tris•HCl pH 7, 300 mM NaCl, 10% (w/v) glycerol, 0.001% Tween 20) with Pierce EDTA-free protease inhibitor mini tablets (ThermoFisher Scientific). The resuspension was incubated with 100 µg/mL lysozyme for 10 min at 4°C and sonicated with a Fisher Scientific Sonic Dismembrator sonicator (FB-705) in 50 mL batches at an amplitude of 30 for 20 s, pulsing for 1 s on and 1 s off, followed by a delay period of 20 s for a series of 12 cycles. Crude lysate was clarified at 21,000x*g* for 25 min at 4°C, at which point the supernatant was applied to pre-equilibrated Strep-Tactin XT Superflow high-capacity resin (IBA Lifesciences) and allowed to enter the column by gravity flow; the streptactin resin had been equilibrated in Strep Wash Buffer (50 mM Tris•HCl pH 7, 300 mM NaCl). The column was washed with 30 column volumes of Strep Wash Buffer and the protein was eluted with 25

column volumes of Strep Elution Buffer (Strep Wash Buffer with 50 mM biotin). Protein eluted from the streptactin column was then applied to pre-equilibrated TALON Metal Affinity Resin (Takara) and allowed to enter the column by gravity flow; the TALON resin was equilibrated in His Wash 1 (50 mM Tris•HCl pH 7, 150 mM NaCl, 5 mM imidazole). A series of 15 column volume washes were applied (His Wash 1; His Wash 2: 50 mM Tris•HCl pH 7, 150 NaCl, 10 mM imidazole) and the protein was eluted in 20 column volumes of His Elution Buffer (50 mM Tris•HCl pH 7, 150 mM NaCl, 150 mM imidazole). The eluted protein was dialyzed overnight in Dialysis Buffer (50 mM Tris•HCl, pH 7, 150 mM NaCl). The next day, the protein was concentrated using a 10,000 NMWL Amicon Ultra-15 Centrifugal Filter Unit at 4°C and the protein concentration was determined by the Bradford assay. ceTIR was flash frozen in liquid nitrogen and stored at $-80$°C in 25 µL aliquots.

For recombinant ceSAMTIR, the tandem SAM domains and the TIR domain of *C. elegans* TIR-1a (residues 557–930) were cloned into the pMAL-c2X vector using PCR-based methods and constructs were propagated in chemically competent XL1-Blue *E. coli* (Agilent). The final construct has an N-terminal MBP tag and Factor-Xa cleavage site. Single colonies were grown overnight in LB with 100 µg/mL ampicillin (final concentration). Plasmid DNA was mini-prepped from the cultures (Promega) and the construct sequence was verified by Sanger sequencing (Genewiz). To express recombinant ceSAMTIR, the ceSAMTIR construct was transformed into chemically competent BL21(DE3) cells and stored as a glycerol stock at $-80$°C. Starter cultures were prepared using an inoculation loop to transfer the transformed bacteria to 5 mL LB with 100 µg/mL ampicillin (final concentration); the culture was grown overnight at 37°C while shaking. The following day, cultures were diluted 1:400 in LB containing 100 µg/mL ampicillin (final concentration) and grown at 37°C while shaking at 215 rpm until an $OD_{600} = 0.5$–0.6 was reached. The cultures were cooled on ice before inducing with 300 µM IPTG (final concentration) and incubating at 16°C for an additional 16–18 hr. Bacteria were pelleted by centrifugation at 3000x*g* for 15 min at 4°C, flash frozen in liquid nitrogen, and stored at $-80$°C until use.

To purify recombinant ceSAMTIR, cell pellets were thawed on ice and resuspended in lysis buffer (50 mM Tris•HCl pH 8, 500 mM NaCl, 1 mM TCEP, 10% glycerol, 0.001% Tween 20) with Pierce EDTA-free protease inhibitor mini tablets (ThermoFisher Scientific). Cells were lysed by incubating the resuspension with 100 µg/mL lysozyme (final concentration) at 4°C for 10 min and sonicating with a Fisher Scientific Sonic Dismembrator sonicator (FB-705) in 50 mL batches at an amplitude of 30 for 20 s, pulsing for 1 s on and 1 s off, followed by a delay period of 20 s for a series of 12 cycles. Lysate was clarified at 17,000x*g* for 25 min at 4°C. The clarified lysate was applied to pre-equilibrated amylose resin (NEB) and purified by gravity flow. Resin was washed with 30 column volumes of wash buffer (50 mM Tris•HCl pH 8, 500 mM NaCl, 1 mM TCEP) and protein was eluted with 10 column volumes of elution buffer (50 mM Tris•HCl pH 8, 500 mM NaCl, 1 mM TCEP, 10 mM maltose). Eluted protein was dialyzed overnight at 4°C in dialysis buffer (50 mM Tris•HCl pH 8, 75 mM NaCl, 0.5 mM TCEP). The next day, the protein was concentrated using a 30,000 NMWL Amicon Ultra-15 centrifugal filter at 4°C. Protein concentration was determined by the Bradford assay. ceSAMTIR was flash frozen in liquid nitrogen and stored at $-80$°C in 25 µL aliquots.

## Determination of SARM1 concentration in lysates

The concentration of SARM1 in the lysates was determined by quantitative western blotting as previously described (*Loring et al., 2020a*). Briefly, serial dilutions of purified TIR domain protein (0–2 µM) were separated by SDS–PAGE along with lysate samples (1: 40 dilutions in duplicate). Next, proteins were transferred to nitrocellulose for western blotting. Protein was detected using Streptavidin conjugated to an IR dye, which recognizes the N-terminal strep-tag. Band intensities were quantified (Licor) and used to generate standard curves, which were then applied to establish the concentration of SARM1 in the lysates.

## Fluorescent assay

We applied a previously described continuous fluorescent assay to monitor SARM1 activity (*Loring et al., 2020a*). This assay employs an $NAD^+$ analog, Nicotinamide 1, $N^6$-ethenoadenine dinucleotide (ENAD), as a substrate. Upon hydrolysis and release of nicotinamide, etheno-ADPR (EADPR) fluoresces ($\lambda_{ex}$=330 nm, $\lambda_{em}$=405 nm). Enzymatic reactions were performed in Assay Buffer (20 mM HEPES pH 8.0 with 150 mM NaCl) and initiated by the addition of a 10x stock solution of ENAD in

96-well Corning Half Area Black Flat Bottom Polystrene NBS plates for a final reaction volume of 60 μL. EADPR fluorescence ($\lambda_{ex}$=330 nm, $\lambda_{em}$=405 nm) was detected in real time at 15 s intervals for the respective time period using a PerkinElmer EnVision 2104 Multilabel Reader in conjunction with Wallac EnVision Manager software. Activity was linear with respect to time and enzyme concentration under the conditions used. EADPR fluorescence ($\lambda_{ex}$=330 nm, $\lambda_{em}$=405 nm) was converted to molarity using an EADPR standard curve. Briefly, fixed concentrations of ENAD (0–400 μM) were treated with excess ADP-ribosyl cyclase (Sigma Aldrich #A9106). The peak fluorescence intensities at each EADPR concentration were plotted against EADPR concentration to generate a standard curve.

Using this assay, the activity was monitored at each stage of the purification: crude lysate, clarified lysate, and purified protein, by adding SARM1 (300 nM) to Assay Buffer and then initiating the reaction with 100 μM ENAD. Activity was also measured by adding SARM1 (300 nM) back to empty pET30a+ vector C43 (DE3) lysate in Assay Buffer with 100 μM ENAD. These assays were performed for 15 min with readings taken every 15 s. The concentration dependence of purified SARM1 was also performed using this assay. The activity was monitored for serial dilutions of SARM1 (5–35 μM) in Assay Buffer after the addition of 1 mM ENAD. Readings were taken at 15 s intervals for 30 min.

## Analytical ultracentrifugation

Analytical ultracentrifugation sedimentation velocity analysis was performed at the UCONN Biophysics core. For these experiments, SARM1 TIR domain was studied at an $OD_{280}$ of 0.3 and 1.0 in triplicate in Assay Buffer at 20℃ and 50,000 rpm using absorbance optics with a Beckman–Coulter Optima analytical ultracentrifuge. Double sector cells equipped with quartz windows were used for analysis. The rotor was equilibrated under vacuum at 20℃ and after a period of ~1 hr at 20℃, the rotor was accelerated to 50,000 rpm. Absorbance scans at 280 nm were acquired at 20 s intervals for ~36 hr. Data was analyzed with the Sedfit program using the direct boundary modeling program for individual data sets and model-based numerical solutions to the Lamm equation. Continuous sedimentation coefficient c(s) distribution plots are sharpened, relative to other analysis methods, because the broadening effects of diffusion are removed by use of an average value for the frictional coefficient. The c(s) analyses were done at a resolution of 0.05 s, using maximum entropy regularization with a 95% confidence limit.

## Effect of crowding agents on activity

Stock solutions of macroviscogens (PEG 8000, 3500, 1500, 400, Dextran) and microviscogens (Sucrose and Glycerol) were prepared at 50% w/v for the PEGs and 60% w/v for the remaining viscogens (Dextran, Sucrose, Glycerol). SARM1 was immersed in Assay Buffer in triplicate at 10 μM without additive or with the addition of 30% w/v of each viscogen. After a 5-min incubation at RT, the reaction was initiated by the addition of 1 mM ENAD. The increase in fluorescence was monitored for 30 min as described above. The fluorescence was converted to EADPR produced using the EADPR standard curve. The slopes of these progress curves yielded the velocity of the reaction. The fold change relative to the no additive control was determined and the log fold change was plotted in GraphPad Prism. A representative progress curve is shown for SARM1 (20 μM) in Assay Buffer with and without 30% w/v PEG 3350. The reaction was initiated with 2 mM ENAD and readings were taken every 15 s for 15 min.

For the macroviscogens and glycerol, a dose dependence study was performed. SARM1 (10 μM) was immersed in Assay Buffer with 0, 10, 20, or 30% w/v of viscogen (PEG 8000, PEG 3350, PEG 1500, PEG 400, dextran, and glycerol) in triplicate. After a 5 min incubation at RT, ENAD was added (1 mM final) to initiate the reaction. The increase in fluorescence produced was monitored for 30 min and converted to velocity values at each viscogen concentration as described above. The fold change relative to the no additive control was determined and plotted in GraphPad Prism.

## Determination of relative viscosities

Viscosities were determined using a falling ball viscometer (Gilmont) and recording the time it takes for a steel ball to fall from the top line to the bottom line when the viscometer is filled with the respective solution. The viscosities were determined for water and 20% w/v solutions of dextran,

sucrose, glycerol, and PEGs 8000, 3350, 1500, and 400 in duplicate. The relative viscosity was determined by calculating the fold-change relative to the viscosity of water.

## Effect of PEG 3350 on enzyme concentration dependence

The enzyme concentration dependence studies were performed in Assay Buffer in the absence of PEG 3350 from 5 to 35 µM of SARM1 and in the presence of 12.5–25% w/v PEG 3350 with 300 nM–20 µM of SARM1 in triplicate. SARM1 was incubated under these conditions for 5 min at RT prior to initiation of the reaction by the addition of ENAD (1 mM). Fluorescence over time was recorded for 30 min and converted to velocity using a standard curve, as described above. Data was plotted in GraphPad Prism and fit to *Equation 1*,

$$Y = B_{max} * X^h / (K_{d^h} + X^h), \tag{1}$$

where, X is the concentration, Y is specific binding, $B_{max}$ is maximum binding, $K_d$ binding coefficient, and h is the hill slope.

## Inhibitory studies

To confirm that SARM1 behaves similarly in the presence and absence of viscogen, we assessed whether divalent metals affect SARM1 activity in the presence of 25% PEG 3350. Activity was recorded with and without 2 mM metals for purified SARM1 (2.5 µM) in Assay Buffer with 25% PEG 3350 after addition of 500 µM ENAD. The activity was also recorded for SARM1 (300 nM) in lysates with 2 mM metals after the addition of 50 µM ENAD. These experiments were performed in duplicate. The activity relative to the no metal control was determined and values plotted in GraphPad Prism. For $CdCl_2$ and $CuCl_2$, $IC_{50}$ values were determined at 500 nM, 1 µM, and 2.5 µM of SARM1 in Assay Buffer with 25% PEG 3350 and 500 µM ENAD.

We next determined $IC_{50}$ values for zinc chloride and berberine chloride in the presence and absence of 25% w/v PEG3350. For the experiments with PEG, SARM1 (2.5 µM) was combined with 25% PEG 3350 in Assay Buffer in triplicate. SARM1 (20 µM) was studied in Assay Buffer for the experiments without PEG 3350. Inhibitor (zinc chloride or berberine chloride) was added at concentrations from 0 to 500 µM and incubated for 5 min at RT prior to initiating the reaction with ENAD (100 µM final). The data were converted to velocity as described above and then normalized to the activity without the inhibitor. $IC_{50}$ values were calculated by fitting the normalized inhibition data to *Equation 2*,

$$\text{Fractional activity of SARM1} = 1/(1 + [I]/IC_{50}) \tag{2}$$

in GraphPad Prism. [I] is the concentration of inhibitor, $IC_{50}$ is the concentration of the inhibitor at half the maximum enzymatic activity, and fractional activity of SARM1 is the percent activity at the respective inhibitor concentration.

To further assess whether the results in lysate are recapitulated with pure protein in the presence of viscogen, mechanism of inhibition studies were repeated with zinc chloride and berberine chloride. For these experiments, SARM1 (2.5 µM) was combined with 25% w/v PEG 3350 in Assay Buffer with either zinc chloride (0–2.5 µM) or berberine chloride (0–200 µM) in triplicate. Reactions were initiated with ENAD (0–2 mM). Reactions were monitored for 30 min as described above. The initial velocities were compiled to produce Michaelis Menten curves at each inhibitor concentration and fit to equations for competitive (*Equation 3*), noncompetitive (*Equation 4*), and uncompetitive inhibition (*Equation 5*),

$$v = V_{max}[S]/[[S] + K_m(1 + [I]/K_{is})] \tag{3}$$

$$v = V_{max}[S]/[[S](1 + [I]/K_i) + K_m(1 + [I]/K_i)] \tag{4}$$

$$v = V_{max}[S]/[[S](1 + [I]/K_{ii}) + K_m] \tag{5}$$

in GraphPad Prism. $K_{ii}$ is the intercept $K_i$ and $K_{is}$ is the slope $K_i$. The best fits were determined through a combination of visual and quantitative analysis.

## PEG 3350 and sodium citrate precipitate SARM1

SARM1 (20 µM) was incubated at RT for 10 min in Assay Buffer with or without 25% PEG 3350 in triplicate. At which time, the solutions were separated by centrifugation at 21,000 x g for 10 min at 4°C. The supernatant was removed, and the pellet resuspended in Assay Buffer plus 25% PEG 3350. The pre-centrifugation, supernatant, and resuspended pellet samples were assayed by the addition of 1 mM ENAD and monitored for 30 min as described above. Coomassie gels were run on aliquots of each of the fractions. The same experiment was also performed at 5, 10, and 20 µM SARM1 in the presence of 0, 10, 17.5, and 25% PEG 3350 in assay buffer. Briefly, SARM1 (5, 10, and 20 µM) was incubated in Assay Buffer containing (0, 10, 17.5, or 25%) PEG 3350 for 10 min at RT. Then the solutions were separated by centrifugation at 21,000 x g for 10 min at 4°C. The supernatant was removed, and pellet resuspended in Assay Buffer plus the respective PEG concentration. Each fraction was assayed for activity via the addition of 1 mM ENAD and presence of SARM1 confirmed by SDS–PAGE and Commassie staining.

These centrifugation experiments were also performed with sodium citrate at concentrations of 100, 250, 500, 750, and 1000 mM in Assay Buffer in triplicate. Specifically, SARM1 (5 µM) was incubated at RT for 10 min in Assay Buffer with different concentrations of sodium citrate. After which, the samples were separated by centrifugation at 21,000 x g for 10 min at 4°C. The supernatant was removed, and the pellet resuspended in Assay Buffer plus sodium citrate. The resuspended pellet samples were assayed via the addition of 1 mM ENAD and monitored for 30 min as described above. The pellet and supernatant samples were separated by SDS–PAGE for each sodium citrate concentration.

## Effect of 1,6–hexanediol on activity

The effect of 1,6–hexanediol, an aliphatic alcohol, on SARM1 activity was assessed. SARM1 (20 µM) was added to Assay Buffer containing 0–2% 1,6–hexanediol and the reaction was initiated via the addition of 1 mM ENAD in triplicate. The reaction was monitored for 1 hr as described above. The effect of 1,6–hexanediol was also assessed in the presence of precipitants. SARM1 (2.5 µM) was added to Assay Buffer with 25% PEG 3350 or 500 mM sodium citrate with and without 2% 1,6–hexanediol in triplicate. The reaction was initiated via the addition of 1 mM ENAD and monitored for 30 min as described above.

## Determination if precipitation is reversible

To investigate whether SARM1 precipitation is reversible, SARM1 (10 µM) was added to Assay Buffer with either 25% PEG 3350 or 500 mM sodium citrate. Samples were then separated by centrifugation at 21,000 x *g* for 10 min at 4°C. The supernatant was removed, and the pellet was resuspended in buffer alone or buffer plus additive. The activity of the different fractions was assessed at 5 µM SARM1 final and 500 µM ENAD as described above. To confirm that there was not a precipitate present after resuspension in buffer alone but still present after resuspension with buffer plus additive, samples were again separated by centrifugation at 21,000 x *g* for 10 min at 4°C and visually inspected. This experiment was performed in triplicate.

## Effect of pH on SARM1 activity

To determine if pH affects the activity of pure enzyme, buffer stocks (4x) were prepared at 80 mM and 600 mM NaCl. The following buffers were used: Sodium Acetate pH 4.0, 4.5, 5.0, and 5.5; Bis–tris pH 6.0 and 6.5; Tris pH 7.0, 7.5, 8.0 and 8.5; and CHES pH 9.0. For the assays, the 4x buffer stocks were diluted with water and 25% PEG 3350 for a final concentration of 20 mM and 150 mM NaCl and the pH was confirmed after dilution. SARM1 (2.5 µM) was added to the buffer plus PEG solution and incubated at RT for 5 min prior to initiation of the reaction with ENAD (0–2 mM). These experiments were performed three times in duplicate. The fluorescence was recorded as described above for 1 hr. The initial velocities were compiled in GraphPad Prism and data for each pH was fit to the Michaelis Menten equation (*Equation 6*),

$$v = V_{max}[S]/(K_m + [S]) \tag{6}$$

$V_{max}$ is the maximum velocity, [S] is the concentration of substrate, and $K_m$ is the substrate

concentration at half the maximum velocity. The log $K_m$, $k_{cat}$ and $k_{cat}/K_m$ values were plotted against pH and fit to a bell-shaped equation (*Equation 7*),

$$Y = Dip + [(Plateau1\_Dip)/(1 + 10\hat{}((LogEC50\_1 - -X) * nH1))] + [(Plateau2\_Dip)/ \\ (1 + 10\hat{}((X - -LogEC50\_2) * nH2))]$$

(7)

X is the Log pH, Y is the response, Plateau 1 and 2 are the initial and final plateaus, Dip is the plateau level between phases, Log EC50_1 and 2 are equivalent to pka1 and 2, and nH1 and 2 are the slope factors.

Percent precipitation was also assessed from pH 4–9 for SARM1 with and without PEG 3350. SARM1 (2.5 µM) in 1x buffer alone or with 25% PEG 3350 was incubated at RT for 10 min. At which time, the samples were separated by centrifugation at 21,000 x g for 10 min at 4°C. The supernatant was removed, and the pellet resuspended in buffer alone or with 25% PEG 3350. Supernatant and pellet samples with and without PEG 3350 were run on an SDS–PAGE gel. Stain-free images were obtained, and bands were quantified using ImageJ software. The percent precipitation values were averaged for the experiment performed in duplicate and the values fit to *Equation 7*.

## Effect of additives on precipitation

SARM1 (20 µM) was immersed in buffer alone (50 mM HEPES pH 8.0 plus 150 mM NaCl) or buffer plus additive (PEG 25% or citrate 500 mM). Zn (10 µM), NMN (100 µM), Ca (100 µM), and berberine chloride (100 µM) were independently incubated with SARM1 plus additive for 20 min prior to separation by centrifugation for 10 min at 21,000 x g at 4 °C. The supernatant was removed and added to SDS–PAGE loading buffer, and the pellet resuspended in SDS–PAGE loading buffer. Supernatant and pellet samples were then run on an SDS–PAGE gel. Stain free images were obtained, and bands were quantified using ImageJ software. This experiment was performed in duplicate.

## Citrate increases the activity of ceSARM1

Using the Fluorescent Assay (see above), the activities of ceTIR and ceSAMTIR were monitored in the absence and presence of 1000 mM sodium citrate. ceTIR or ceSAMTIR at 2.5 µM (final concentration) and 1000 mM sodium citrate (final concentration) were added to assay buffer (50 mM Tris•HCl pH 8, 150 mM NaCl). Following a brief 10 min equilibration period at room temperature, the reaction was initiated by the addition of 1 mM ENAD (final concentration). Fluorescence intensity readings ($\lambda_{ex}$ = 330 nm, $\lambda_{em}$ = 405 nm) were taken every 15 s for 17.5 min at 25°C and the increase in fluorescence was converted to velocity; the fluorescence intensity readings were converted to [EADPR] using the EADPR standard curve (discussed above) and the slopes of the resulting progress curves yielded the velocity. The fold change in velocity relative to the no citrate control was plotted in GraphPad Prism.

## Dose response of ceSAMTIR precipitation and associated activity in sodium citrate

ceSAMTIR (10 µM) was incubated in assay buffer (50 mM Tris•HCl pH 8, 150 mM NaCl) containing 0, 125, 250, 500, 750, or 1000 mM sodium citrate for 15 min at room temperature. Precentrifugation controls were removed, at which point, the samples were centrifuged at 21,000xg for 10 min at 4°C. The supernatant fraction was removed, and the pellet was resuspended in assay buffer containing the corresponding sodium citrate concentration. The precentrifugation, supernatant, and pellet fractions were analyzed by SDS-PAGE and Coomassie staining for the presence of ceSAMTIR. All fractions were also assayed for enzymatic activity in duplicate using the Fluorescent Assay with 1 mM ENAD. Fluorescence increase over time was converted to velocity and plotted using GraphPad Prism.

## Negative stain electron microscopy on SARM1

We performed negative stain EM on SARM1 TIR domain (10 µg/mL) in 50 mM HEPES pH 8.0 plus 150 mM NaCl with and without sodium citrate (500 mM). Samples were processed by the core facility at UMass Medical School. Samples were fixed with uranyl acetate and imaged on the Philips CM120 microscope. Particle areas (n = 100) were analyzed in ImageJ and graphed in GraphPad Prism. Significance was determined by a student's T–test (p value = 5.4E-42).

## Effect of PEG 3350 on steady state kinetics

The steady state kinetic reactions were performed in Assay Buffer with varied PEG 3350 (0–25%) at a constant SARM1 concentration (10 µM) in triplicate. Reactions were incubated at RT for 5 min and initiated by the addition of ENAD (0–2 mM final). Fluorescence was recorded as described above for 1 hr. The initial velocities were compiled and fit to *Equation 6* in GraphPad Prism to produce Michaelis Menten curves at each PEG 3350 concentration.

The fold change was calculated for the $K_m$, $k_{cat}$, and $k_{cat}/K_m$ values at 10 µM SARM1 with the addition of 25% w/v PEG 3350 compared to the values without PEG 3350 and the log values plotted using GraphPad Prism. The $K_m$, $k_{cat}$ and $k_{cat}/K_m$ values were also plotted against PEG 3350 concentration and trends depicted with spline ($K_m$) or sigmoidal fits ($k_{cat}$ and $k_{cat}/K_m$) (*Equation 8*),

$$Y = \text{Bottom} + (\text{Top} - \text{Bottom})/(1 + 10^{((\text{LogIC50} - X)^* \text{HillSlope})}) \tag{8}$$

X is the log(concentration), Y is the response, Top and Bottom are plateaus, log $IC_{50}$ is the log of the concentration of the inhibitor at half the maximum enzymatic activity.

These experiments were also performed at constant PEG 3350 (25%) and varied SARM1 concentration (500 nM, 1 µM, 2.5 µM, 5 µM, and 10 µM), and without PEG 3350 at 10 and 20 µM SARM1 in triplicate. SARM1 was incubated in Assay Buffer with or without PEG 3350 at RT for 5 min and then the reaction was initiated by the addition of ENAD (0–2 mM final). Fluorescence was recorded as described above for 1 hr. The initial velocities were compiled to produce Michaelis Menten curves at each SARM1 concentration with and without PEG 3350. These slopes were plotted in GraphPad Prism and fit to the Michaelis Menten equation (*Equation 6*). From this analysis, the $K_m$, $k_{cat}$ and $k_{cat}/K_m$ values were plotted against enzyme concentration and the trend is shown with a spline fit.

## LC–MS-based time course experiments

ADPR, cADPR, nicotinamide, and NAD$^+$ standards were prepared at 100 µM in Assay Buffer. Purified SARM1 (20 µM) was incubated with 100 µM NAD$^+$ in Assay Buffer and the reaction was monitored over time. Samples were injected at 0, 15, 40, 90, and 120 min onto an InfinityLab Poroshell 120 EC–18 column (4.6 x 50 mm, 2.7 mircron, Agilent #699975–902T) at a flow rate of 0.4 mL/min. Metabolites were eluted with a gradient of 100% H$_2$O with 0.01% formic acid from 0 to 6 min, 90% H$_2$O with 0.01% formic acid and 10% acetonitrile with 0.1% formic acid from 12 to 14 min, and then 100% H$_2$O with 0.01% formic acid from 14 to 20 min. The metabolites were detected with a quadrupole mass spectrometer (Agilent, G6120B single quad, ESI source, 1260 Infinity HPLC) in the positive ion mode. The absorbance at 254 nm was plotted over time at each reaction time point in GraphPad Prism. Reaction time courses were also performed as described above, in the presence of 25% PEG 3350 or 500 mM sodium citrate at 2.5 µM SARM1. The reactions were injected at time 0, 5, 15, 30, 60, 90, and 120 min for PEG 3350 and 0, 10, 30, 60, 90, 120, and 150 min for sodium citrate. Peak areas were quantified at an absorbance of 254 nm and their accumulation or consumption plotted over time. The ratio of ADPR to cADPR was also quantified for each condition over time and averaged for the entire time course. A student's t-test was performed to determine if the differences in ADPR to cADPR ratios in the absence or presence of additive were statistically significant (p value = 0.00036 for No PEG 3350/PEG 3350, 0.0000024 No sodium citrate/sodium citrate, and 0.000014) for PEG 3350/sodium citrate.

Reaction time courses were also performed with 100 µM cADPR and SARM1 alone (20 µM), and with 25% PEG 3350 or 500 mM sodium citrate at 5 µM enzyme in Assay Buffer. For SARM1 alone and SARM1 with sodium citrate, the reaction was monitored at time 0, 2 hr and 6 hr. SARM1 with PEG 3350 was recorded at time 0, 10, 30, 60, 90, and 150 min. Samples at each time point were injected onto an InfinityLab Poroshell 120 EC–18 column (4.6 x 50 mm, 2.7 mircron, Agilent #699975–902T) at a flow rate of 0.4 mL/min. Metabolites were eluted with a gradient of 100% H$_2$O with 0.01% formic acid from 0 to 6 min, 90% H$_2$O with 0.01% formic acid and 10% acetonitrile with 0.1% formic acid from 12 to 14 min, and then 100% H$_2$O with 0.01% formic acid from 14 to 20 min. The metabolites were detected with a quadrupole mass spectrometer (Agilent, G6120B single quad, ESI source, 1260 Infinity HPLC) in the positive ion mode. The absorbance at 254 nm was plotted over time at each reaction time point in GraphPad Prism. The peak areas were quantified for ADPR and cADPR at an absorbance of 254 nm and the average was over time was plotted in GraphPad Prism. These experiments were performed in triplicate.

### Single turnover experiment

SARM1 (20 µM) was treated with 10 µM NAD$^+$ in 50 mM HEPES pH 8.0, 150 mM NaCl in duplicate and reactions were injected on to an InfinityLab Poroshell 120 EC–18 column (4.6 x 50 mm, 2.7 mircron, Agilent #699975–902T) at a flow rate of 0.4 mL/min at 0, 5, 30, and 60 min time points. Metabolites were eluted with a gradient of 100% H$_2$O with 0.01% formic acid from 0 to 6 min, 90% H$_2$O with 0.01% formic acid and 10% acetonitrile with 0.1% formic acid from 12 to 14 min, and then 100% H$_2$O with 0.01% formic acid from 14 to 20 min. The metabolites were detected with a quadrupole mass spectrometer (Agilent, G6120B single quad, ESI source, 1260 Infinity HPLC) in the positive ion mode. The absorbance at 254 nm was plotted over time at each reaction time point in GraphPad Prism. The peak areas were quantified for ADPR and cADPR at an absorbance of 254 nm and the average ratio was over time was plotted in GraphPad Prism.

### Inhibition and mechanism of cADPR

SARM1 (2.5 µM) was treated with cADPR (0–1 mM) in the presence of 500 µM ENAD in 50 mM HEPES pH 8.0 with 150 mM NaCl and 25% PEG 3350 in triplicate. The fluorescence over time was monitored for 30 min as described above. The data were converted to velocity as described above and then normalized to the activity without the inhibitor. IC$_{50}$ values were calculated by fitting the normalized inhibition data to *Equation 2*. To determine the mechanism of inhibition cADPR was varied (0, 50, 100, and 200 µM) at 0–400 µM ENAD at 2.5 µM SARM1 in 50 mM HEPES pH 8.0 with 150 mM NaCl and 25% PEG 3350 in duplicate. The fluorescence over time was monitored for 30 min as described above. The initial velocities were compiled to produce Michaelis Menten curves at each inhibitor concentration and fit to *Equation 3* for competitive inhibition.

### Substrate specificity

To evaluate the cleavage of 16 potential SARM1 substrates, SARM1 was incubated with and without PEG in 50 mM HEPES pH 8.0 with 150 mM NaCl for 30 min in the presence of 25% PEG and 1 hr in the absence in duplicate. Substrates were analyzed at 100 µM and SARM1 was incubated at a concentration of 2.5 µM in the presence of PEG and 20 µM in its absence. No enzyme controls were run for each of the substrates in the absence and presence of PEG 3350. Samples were injected onto an InfinityLab Poroshell 120 EC–18 column (4.6 x 50 mm, 2.7 micron, Agilent #699975–902T) at a flow rate of 0.4 mL/min. Note that the flow rate was slowed down from 0.4 mL/min to 0.15 mL/min for 0–6 min for FAD, ATP, ADP, and GTP. Metabolites were eluted with a gradient of 100% H$_2$O with 0.01% formic acid from 0 to 6 min, 90% H$_2$O with 0.01% formic acid and 10% acetonitrile with 0.1% formic acid from 12 to 14 min, and then 100% H$_2$O with 0.01% formic acid from 14 to 20 min. The metabolites were detected with a quadrupole mass spectrometer (Agilent, G6120B single quad, ESI source, 1260 Infinity HPLC) in the positive ion mode. Percent cleavage was calculated by quantification of peak areas at an absorbance of 254 nm for products divided by that of the products plus substrate times 100%.

### TIR domain mutants

TIR domain mutants were generated using the pET30a+ TIR domain construct as a template (Appendix 1). To mutagenize the plasmid by PCR, template DNA (2 ng/µL) was combined with forward and reverse primers (0.5 µM final, Appendix 1) in 1xNEBNext Mastermix, which contains the Q5 high fidelity polymerase, deoxynucleotides, and 2 mM MgCl$_2$. The PCR protocol was as follows: (1) initial denaturation at 98°C for 1 min, (2) denaturation at 98°C for 30 s, (3) annealing at 54–56°C for 45 s, (4) extension at 68°C for 1 min and (5) final extension at 68°C for 5 min. Steps 2–4 were repeated 30 times. Site-directed mutagenesis was confirmed by Sanger sequencing (Genewiz).

Steady state kinetic parameters were determined at 5 µM mutant SARM1 and 25% PEG 3350 in Assay Buffer in duplicate. Reactions were initiated by the addition of ENAD (0–2 mM). Fluorescence was recorded as described above for 1 hr. The initial velocities were compiled in GraphPad Prism and fit to the Michaelis Menten equation (*Equation 6*). From this analysis, the $K_m$, $k_{cat}$, and $k_{cat}/K_m$ values were obtained for each mutant and plotted in GraphPad Prism.

Precipitation of mutants was assessed by combining SARM1 (10 µM) with 25% PEG 3350 in Assay Buffer in duplicate. Samples were separated by centrifugation at 21,000 x g for 10 min at 4°C. The supernatant was removed, and the pellet was resuspended in buffer. Proteins were separated by

SDS–PAGE in duplicate and band intensities were quantified with ImageJ and percent precipitation plotted in GraphPad Prism.

## Creation of mammalian GFP SAMTIR mutants

A GFP-tagged SARM1 SAMTIR DNA construct (Residues 412–724, Uniprot) in a pcDNA3.1(+) vector was ordered by custom synthesis from GenScript (Appendix 1). Site directed mutagenesis was performed as described above for the TIR domain mutants with the primers listed in Appendix 1. Mutations were confirmed by Genewiz Sanger Sequencing using T7 and BGHR primers.

## Transient transfection of HEK293T cells

Mycoplasma negative HEK293T cells purchased from the ATCC were grown to 60% confluency prior to transfection. Cells tested negative for mycoplasma contamination. 10 µg of E642A and E642A/G601P GFP-tagged SAMTIR DNA was diluted into 500 µL of OPTIMEM Reduced serum media. At the same time, 30 µL of lipofectamine 2000 was added to 470 µL of OPTIMEM Reduced serum media and incubated for 3–5 min at RT. Next, the DNA mix was added to the lipofectamine mix and this solution was incubated for 20 min at RT. At which point, 9 mL of OPTIMEM was added to the DNA lipofectamine mix and applied to replace the cell culture media. After a 6 hr incubation at 37°C in a $CO_2$ incubator, the media was removed and replaced with fresh DMEM media with 10% FBS and 1% Pen/Strep. The cells were incubated for another 18 hr prior to collection for FACS.

## Fluorescence-activated cell sorting

E642A and E642A/G601P GFP–SAMTIR cells were collected by washing with PBS, and then adding 3 mL trypsin for 3 min and quenching the reaction with 7 mL DMEM. Cells were resuspended in media and spun down at 3000 x*g* for 3 min to remove the trypsin. Cells were next resuspended in PBS with 1% FBS for FACS. 7AAD (1 µL) was added for a 30 min incubation on ice to stain the dead cells prior to FACS. Cells were sorted by the flow cytometry core at UMass Medical School. Specifically, GFP-positive 7AAD-negative singlet transfected cells were sorted via 85 micron, 47.5 frequency, four–way purity on the BD FACS Aria II Cell Sorter with a purity mask of 32, plates voltage of 4500, and sheath pressure of 45. Cells were collected in PBS and then imaged using Amnis Flow-Sight Imaging Cytometer.

## Amnis FlowSight imaging studies

Images were acquired by the Flow Cytometry Core at UMass Medical School using the Amnis Flow-Sight Imaging Cytometer (EMD Millipore) with IDEAS software (EMD Millipore). 20,927 images were acquired for E642A and 9895 images were obtained for E642A/G601P GFP–tagged SAMTIR. Channel 1 (430–480 nm) and channel 2 (505–560 nm) images were obtained for bright field and GFP, respectively. All cells were GFP positive and 7AAD negative as expected due to the FACS selection. Gates were applied to prioritize images that were in focus and taken of cells as singlets, and the intensity of these were obtained (*Figure 6—figure supplement 2*). These cells were analyzed by the IDEAS spot count wizard to bucket the cells according to number of puncta. R1 and R2 designate distinct subpopulations where R1 resembles baseline fluorescence and R2 has higher puncta than baseline.

## *C. elegans* strains and culture

Unless otherwise noted, all *C. elegans* strains were grown and maintained at 20°C on standard nematode growth media (NGM) plates seeded with *Escherichia coli* OP50 (*Sulston and Hodgkin, 1988*). Strain ZD101 (*tir–1(qd4)*) was provided by the Caenorhabditis Genetics Center, which is funded by NIH Office of Research Infrastructure Programs (P40 OD010440). Strains analyzed: ABC346 (*oxIs12 [unc–47 p::GFP, lin–15(+)] X; bamEx1008 [unc–47 p::mScarlet::tir–1b::let–858 3'UTR (25 ng/µl) +ceh–22::gfp (15 ng/µl)]*) and ABC16 (*tir–1(qd4);oxIs12*).

## Citrate treatment of *C. elegans*

Animals were exposed to citrate by placing them on NGM plates containing 8 mM sodium citrate and seeded with *E. coli* OP50. Fourth larval stage (L4) animals were placed on citrate plates or standard NGM plates.

## Confocal imaging

Animals were immobilized with 0.1 mM levamisole on 3% agarose pads. Animals were imaged on Perkin Elmer Precisely UltraVIEW VoX confocal imaging system using a 40 X objective. Images were compressed and exported as TIFF files for processing in FIJI.

## TIR-1 puncta analysis

TIR–one puncta were analyzed using Fiji software (*Schindelin et al., 2012*). Defined intensity threshold values were acquired from control experiments for each fluorescent marker. The 'analyze particles' function of Fiji was used to determine the number of puncta and the area of each individual puncta in a given region of interest (ROI) along the ventral nerve cord. Cell bodies were excluded from analysis. The mean number of puncta was determined by dividing the total number of puncta in all ROIs by the total number of ROIs analyzed. The mean puncta area represents the total puncta area divided by the number of puncta. Statistical analysis was performed with GraphPad Prism software.

## Damage and TIR-1 localization analysis

Axon integrity was quantified by counting broken, truncated, and absent axons. The percentage of damaged axons was determined by (total # damaged axons)/(total # axons).

## Statistical analysis

Statistical analysis was performed using GraphPad QuickCalcs (https://www.graphpad.com/quickcals/) and GraphPad Prism. Categorical data: Error bars represent 95% confidence intervals and significance was determined with Fisher's exact test, where $*p < 0.05$, $**p < 0.01$, $***p < 0.001$. Continuous data: unless otherwise noted, data are presented as the mean with error bars representing the standard error of the mean. Significance was calculated with T-tests, where $*p < 0.05$, $**p < 0.01$, $***p < 0.001$, $****p < 0.0001$.

## Acknowledgements

Funding Sources: This work was supported in part by NIH grants R35 GM118112 (PRT), F31 NS108610 (HSL), and T32 AI132152 (JDI). The *C. elegans* work was supported in part by NIH R01 NS110936 (ABB).

## Additional information

### Funding

| Funder | Grant reference number | Author |
| --- | --- | --- |
| National Institute of General Medical Sciences | R35 GM118112 | Paul R Thompson |
| National Institute of Neurological Disorders and Stroke | F31 NS108610 | Heather S Loring |
| National Institute of Neurological Disorders and Stroke | R01 NS110936 | Alexandra B Byrne |
| National Institute of Allergy and Infectious Diseases | T32 AI132152 | Janneke D Icso |

The funders had no role in study design, data collection and interpretation, or the decision to submit the work for publication.

### Author contributions

Heather S Loring, Conceptualization, Formal analysis, Investigation, Methodology, Writing - original draft, Writing - review and editing; Victoria L Czech, Alexandra B Byrne, Formal analysis, Investigation, Methodology, Writing - original draft; Janneke D Icso, Investigation, Writing - original draft; Lauren O'Connor, Formal analysis, Investigation, Methodology; Sangram S Parelkar, Investigation,

Methodology; Paul R Thompson, Conceptualization, Data curation, Formal analysis, Supervision, Funding acquisition, Writing - original draft, Project administration, Writing - review and editing

### Author ORCIDs

Alexandra B Byrne (iD) http://orcid.org/0000-0002-7449-9188
Paul R Thompson (iD) https://orcid.org/0000-0002-1621-3372

### Decision letter and Author response

Decision letter https://doi.org/10.7554/eLife.66694.sa1
Author response https://doi.org/10.7554/eLife.66694.sa2

---

## Additional files

### Supplementary files

• Source data 1. Source data corresponding to the panels shown in *Figure 2*, *Figure 3*, *Figure 4*, *Figure 5*, *Figure 6*, *Figure 2—figure supplements 1* and *2*, *Figure 3—figure supplement 1*, *Figure 5—figure supplements 1–4*, and *Figure 6—figure supplement 1*.

• Supplementary file 1. Supplementary Tables. (A) The Steady-State Kinetic Parameters for Pure Wild Type SARM1 TIR (10 µM) with Varied PEG. (B) The Steady-State Kinetic Parameters for Pure Wild Type SARM1 TIR with Varied Enzyme Concentration.

• Transparent reporting form

### Data availability

All data generated or analysed during this study are included in the manuscript and supporting files.

---

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

# Appendix 1

**Appendix 1—key resources table**

| Reagent type (species) or resource | Designation | Source or reference | Identifiers | Additional information |
|---|---|---|---|---|
| Chemical compound, drug | 1,6–Hexanediol | Sigma Aldrich | 240117 | |
| Chemical compound, drug | 3–acetylpyridine adenine dinucleotide | Sigma Aldrich | A5251 | |
| Chemical compound, drug | 7–AAD | Thermo Fisher | A1310 | |
| Chemical compound, drug | ADPR | Sigma Aldrich | A0752 | |
| Recombinant DNA reagent | ADPR cyclase | Sigma Aldrich | A9106, C7344 | |
| Chemical compound, drug | ADP | Sigma Aldrich | A2754 | |
| Chemical compound, drug | ATP | Sigma Aldrich | A1852 | |
| Chemical compound, drug | Berberine chloride | Sigma Aldrich | B–3251; PHR1502 | |
| Cell line (*Escherichia coli*) | C43 (DE3) | Sigma Aldrich | CMC0019 | |
| Chemical compound, drug | Cadmium Chloride | Sigma Aldrich | 202908 | |
| Chemical compound, drug | cADPR | Sigma Aldrich | C7344 | |
| Chemical compound, drug | Calcium Chloride | Sigma Aldrich | C1016 | |
| Chemical compound, drug | Cobalt (II) Chloride | Sigma Aldrich | 60818 | |
| Chemical compound, drug | Copper (II) Chloride | Sigma Aldrich | 203149 | |
| Chemical compound, drug | Dextran | Fisher Scientific | ICN16011010 | |
| Chemical compound, drug | DMEM | Sigma Aldrich | SLM–241 | |
| Chemical compound, drug | EDTA | Sigma Aldrich | E9884 | |
| Chemical compound, drug | FAD | Sigma Aldrich | F6625 | |
| Chemical compound, drug | Glycerol | Sigma Aldrich | G5516 | |
| Chemical compound, drug | GTP | Sigma Aldrich | 10106399001 | |
| Chemical compound, drug | IPTG | Sigma Aldrich | I6758 | |
| Chemical compound, drug | Kanamycin | Research Products International | 25389–94–0 | |
| Chemical compound, drug | Lipofectamine 2000 | Thermo Fisher | 11668027 | |

*Continued on next page*

*Appendix 1—key resources table continued*

| Reagent type (species) or resource | Designation | Source or reference | Identifiers | Additional information |
|---|---|---|---|---|
| Chemical compound, drug | Magnesium Chloride | Sigma Aldrich | M8266 | |
| Chemical compound, drug | Manganese (II) Chloride | Sigma Aldrich | 244589 | |
| Chemical compound, drug | NAD | Sigma Aldrich | N0632 | |
| Chemical compound, drug | NADH | Sigma Aldrich | N4505 | |
| Chemical compound, drug | NADP | Sigma Aldrich | NADP–RO | |
| Commercial assay, kit | NEBNext Master Mix | New England BioLabs | M0541L | |
| Chemical compound, drug | Nickel (II) Chloride | Sigma Aldrich | 339350 | |
| Chemical compound, drug | Nicotinamide | Sigma Aldrich | 72340 | |
| Chemical compound, drug | Nicotinamide $1,N^6-$ethenoadenine dinucleotide $NAD^+$ | Sigma Aldrich Sigma Aldrich | N2630 N0632 | |
| Chemical compound, drug | Nicotinamide hypoxanthine dinucleotide | Sigma Aldrich | N6506 | |
| Chemical compound, drug | Nicotinamide hypoxanthine dinucleotide, reduced form | Sigma Aldrich | N6756 | |
| Chemical compound, drug | Nicotinic acid adenine dinucleotide | Sigma Aldrich | N4256 | |
| Commercial assay, kit | OPTIMEM | Thermo Fisher | 31985062 | |
| Chemical compound, drug | PEG 400 | Sigma Aldrich | 1546445 | |
| Chemical compound, drug | PEG 1500 | Sigma Aldrich | 10783641001 | |
| Chemical compound, drug | PEG 3350 | Sigma Aldrich | 1546547 | |
| Chemical compound, drug | PEG 8000 | Sigma Aldrich | 1546605 | |
| Commercial assay, kit | Pierce EDTA–free protease inhibitor tablets | Thermo Fisher Scientific | A32955 | |
| Chemical compound, drug | Sodium citrate | Sigma Aldrich | W302600 | |
| Commercial assay, kit | Streptacin XT | IBA Life Sciences | 2–4010–025 | |
| Peptide, recombinant protein | IRDye labeled Streptavidin | LI–COR Biosciences | 926–68079 Lot #C60504–02 | IR Dye 680RD (1:20,000) |

*Continued on next page*

*Appendix 1—key resources table continued*

| Reagent type (species) or resource | Designation | Source or reference | Identifiers | Additional information |
|---|---|---|---|---|
| Chemical compound, drug | Thionicotinamide adenine dinucleotide | Sigma Aldrich | T7375 | |
| Chemical compound, drug | Zinc Chloride | Sigma Aldrich | 229997 | |
| Gene | pET30a$^+$ Strep–TIR | *Loring et al., 2020a* | | |
| Recombinant DNA reagent | pcDNA3.1(+) GFP SARM1 SAMTIR | This study | | GFP– GGSG linker – SARM1 SAMTIR, Residues 412–724 (Uniprot) |
| Recombinant DNA reagent | pET30a$^+$ Strep-ceTIR-HIS | *Loring et al., 2020a* | | |
| Recombinant DNA reagent | pMAL-c2X MBP-ceSAMTIR | This study. | | MBP – Factor Xa – TIR-1a SAMTIR, Residues 557–930 (Wormbase) |
| Sequence-based reagent | Y568A For | IDT | GATGTGTTCATCAG CGCGCGTCGTAACAGC | |
| Sequence-based reagent | Y568A Rev | IDT | CTACACAAGTAGTC GCGCGCAGCATTGTCG | |
| Sequence-based reagent | Y568F For | IDT | GATGTGTTCATCAGC TTCCGTCGTAACAGC | |
| Sequence-based reagent | Y568F Rev | IDT | CTACACAAGTAGTCG AAGGCAGCATTGTCG | |
| Sequence-based reagent | D594A For | IDT | GGCTTCAGCGTGTT CATCGCTGTTGAAAAA | |
| Sequence-based reagent | D594A Rev | IDT | CGCCTCCAGTTTTT CAACAGCGATGAACAC | |
| Sequence-based reagent | G601P For | IDT | GAAAAACTGGAGGC GCCAAAGTTCGAGGAC | |
| Sequence-based reagent | G601P Rev | IDT | CTTTTTGACCTCCGC GGTTTCAAGCTCCTG | |
| Sequence-based reagent | D627A For | IDT | CTGAGCCCGGGT GCGCTGGCTAAGTGTATG | |
| Sequence-based reagent | D627A Rev | IDT | GTGGTCCTGCATACA CTTAGCCAGCGCACCCGG | |
| Sequence-based reagent | K628A For | IDT | AGCCCGGGTGCGC TGGATGCGTGTATG | |
| Sequence-based reagent | K628A Rev | IDT | ATCGTGGTCCTGCATA CACGCATCCAGCGC | |
| Sequence-based reagent | C629A For | IDT | GCGCTGGATAAGG CTATGCAGGACCACGAT | |
| Sequence-based reagent | C629A Rev | IDT | CGCGACCTATTCCGA TACGTCCTGGTGCTA | |
| Sequence-based reagent | C629S For | IDT | GCGCTGGATAAGAG CATGCAGGACCACGAT | |
| Sequence-based reagent | C629S Rev | IDT | CGCGACCTATTCTCGT ACGTCCTGGTGCTAACGATGC | |
| Sequence-based reagent | C635A For | IDT | TAAGGACTGGGTGC ACAAAGAAATCG | |

*Continued on next page*

*Appendix 1—key resources table continued*

| Reagent type (species) or resource | Designation | Source or reference | Identifiers | Additional information |
|---|---|---|---|---|
| Sequence-based reagent | C635A Rev | IDT | TCCTTAGCATCGTGGT CCTGCATACACTTATCC | |
| Sequence-based reagent | C635S For | IDT | ATGCAGGACCACGA TAGCAAGGACTGGGTG | |
| Sequence-based reagent | C635S Rev | IDT | TACGTCCTGGTGCT ATCGTTCCTGACCCAC | |
| Sequence-based reagent | W638A For | IDT | GACCACGATTGCA AGGACGCGGTGCACAAA | |
| Sequence-based reagent | W638A Rev | IDT | GGTAACGATTTCTTT GTGCACCGCGTCCTTGCAAT | |
| Sequence-based reagent | H640A For | IDT | GATTGCAAGGACTGG GTGGCTAAAGAAATC | |
| Sequence-based reagent | H640A Rev | IDT | AGCGGTAACGATTT CTTTAGCCACCCA | |
| Sequence-based reagent | E642A For | IDT | TGGGTGCACAAAGC AATCGTTACCGCTCTG | |
| Sequence-based reagent | E642A Rev | IDT | ACCCACGTGTTTCGTT AGCAATGGCGAGAC | |
| Sequence-based reagent | E642Q For | IDT | AAGGACTGGGTGC ACAAACAGATCGTT | |
| Sequence-based reagent | E642Q Rev | IDT | GCTCAGAGCGGTA ACGATCTGTTTGTG | |
| Sequence-based reagent | H685A For | IDT | AACGGTATCAAGT GGAGCGCTGAATACCAG | |
| Sequence-based reagent | H685A Rev | IDT | CGCTTCCTGGTATTC AGCGCTCCA | |
| Sequence-based reagent | E686A For | IDT | GGTATCAAGTGGAGC CACGCTTACCAGGAA | |
| Sequence-based reagent | E686A Rev | IDT | GATGGTCGCTTCC TGGTAAGCGTGGCT | |
| Sequence-based reagent | Q688A For | IDT | AAGTGGAGCCACGA ATACGCTGAAGCGAC | |
| Sequence-based reagent | Q688A Rev | IDT | TTTCTCGATGGTCGC TTCAGCGTATTCGTG | |
| Sequence-based reagent | SAMTIR E642A For | IDT | TGGGTGCATA AAGCAATTGTGA | |
| Sequence-based reagent | SAMTIR E642A Rev | IDT | GCGCGGTCAC AATTGCTTT | |
| Sequence-based reagent | SAMTIR G601P For | IDT | ACTGGAAGCGC CCAAATTTGAA | |
| Sequence-based reagent | SAMTIR G601P Rev | IDT | TTCAAATTTGGGC GCTTCCAGT | |
| Sequence-based reagent | pMAL ceSAMTIR For | IDT | AAAAAAGGATCCATG GTGCCGGGTTGGACC | |
| Sequence-based reagent | pMAL ceSAMTIR Rev | IDT | AAAAAATCTAGATTAGT TACGGTCGCTGGTGGTGC | |

