## [Decision Letter]

**Acceptance summary:**

This manuscript describes an interesting regulatory mechanism that activates SARM1, an enzyme that degrade NAD+ and promote axon degeneration. Previous structural and biochemical studies mostly focus on how SARM1 is auto-inhibited at basal conditions and this manuscript provides evidences supporting that phase transition could promote its activity, thus providing new insights about its in vivo regulatory mechanism. The finding also enables in vitro assays to be carried out more easily and thus could facilitate the development of small molecule modulators of SARM1 for therapeutics purposes.

**Decision letter after peer review:**

Thank you for submitting your article "A Phase Transition Enhances the Catalytic Activity of SARM1" for consideration by *eLife*. Your article has been reviewed by 3 peer reviewers, including Hening Lin as the Reviewing Editor and Reviewer #1, and the evaluation has been overseen by Cynthia Wolberger as the Senior Editor. The following individual involved in review of your submission has agreed to reveal their identity: Charlie Brenner (Reviewer #2).

Essential revisions:

1) Test the phase transition behavior of full length SARM1 (could be either human or worm SARM1) to see whether the phase transition and activity enhancement also happen to full length SARM1. If the result is positive, it will further enhance the manuscript. If the result is negative, please tone down the claim and discuss this accordingly.

*Reviewer #1 (Recommendations for the authors):*

A few key experiments, including the in vitro activation by PEG or citrate, and the puncta formation in HEK293T cells, should be replicated using the full length SARM1. If similar results are obtained with the full length SARM1, the conclusion will be more strongly supported and further enhance the connection to physiology/pathophysiology. Given that the in vivo data is obtained using *C. elegans*, the *C. elegans* full length SARM1 could be used for such study.

The observation that citrate promotes axon degeneration in *C. elegans* is interesting. The manuscript may use a little more discussion on this topic. For example, it would be interesting to discuss whether there are connections between citrate and axon degermation reported in the literature and whether citrate could be a physiological/pathophysiological trigger for axon degeneration in mammals.

*Reviewer #3 (Recommendations for the authors):*

This study of SARM1 builds on past information and extends it into a more complete picture of the conversion between inactive and active enzyme forms. A biological advance comes from the discovery that citrate can activate the enzyme. Citrate is then used in *C. elegans* to cause in vivo activation of the enzyme with subsequent neuronal damage. There is much to appreciate in this work as it advances the understanding of SARM1 activity and its role in neuronal degradation. There are some questions and areas of improvements to be made, summarized in the comments below. Once these are addressed, the manuscript should be published.

An excellent TOC figure, it is worth more than the proverbial 1000 words.

The title needs to include the catalytic function, perhaps "A Phase Transition Enhances the Catalytic Activity of SARM1, an NAD+ glycohydolase'.

Figure 1 legend should explain proposed domain function.

Table 1 and the description "As before, we found that ZnCl2 inhibits SARM1 noncompetitively with a Ki value of 1.0 ± 0.1 μM." Do not appear to match.

"we found that berberine chloride inhibits SARM1 noncompetitively in the presence of PEG 3350 with a Ki value of 120 ± 10 μM." What is it? A Zn-chelator? Why is it being compared to Zn?

Throughout, the units of uM/sec are ambiguous without knowing the enzyme concentration, requiring a search through the methods/legends. In contrast a rate in the traditional enzymatic units of turnovers / sec (simply s^-1^) seem more descriptive and are used in several critical places, eg. Figure 3 l. The reviewer is reluctant to ask the authors to make all the changes, but include as a note for future reports.

Even in its most active form SARM1 has a k_cat_/K_m_ around 103 s^-1^M^-1^s^-1^, making it a sleepy enzyme, the authors should compare with other known NAD glycohydrolases, and comment as to this difference and how much SARM1 there is in neuronal bodies (300 nM is cited) relative to an ability to deplete NAD pools.

“…cADPR inhibits SARM1 in the presence of PEG with an IC_50_ of 100 ± 10 μM and a competitive inhibition pattern…” What is the mechanism? If both cADPR and ADPR are competitive, it supports a rapid-equilibrium random release of products.

The statement "Although the percentage of damaged axons in the citrate treated animals 2-3 h post-treatment is higher than untreated controls the data are not significant (Figure 6I)." is an obvious logical error. Not significant means not significant and the statement should be that there is no significant change in damaged axons at 2-3 h.

The studies with *C. elegans*, citrate activation and axonal damage are a valuable addition. I was waiting for the NAD analysis to follow… – can the authors measure NAD in control vs untreated C.e.?

A comment needs to address the 500 mM citrate effects on SARM1 versus the 8 mM citrate effect on the *C. elegans* homologue of SARM1.

---

## [Author Response]

Essential revisions:1) Test the phase transition behavior of full length SARM1 (could be either human or worm SARM1) to see whether the phase transition and activity enhancement also happen to full length SARM1. If the result is positive, it will further enhance the manuscript. If the result is negative, please tone down the claim and discuss this accordingly.

We thank the reviewer for the suggestion. To address this issue, we used TIR-1, the *Caenorhabditis elegans* ortholog of SARM1. Like SARM1, TIR-1 also possesses NAD^+^ hydrolase activity. Notably, *C. elegans* encodes several TIR-1 splice variants including TIR-1b and TIR-1d which lack the ARM domain (Figure S5A ) but contain the tandem SAM domains essential for octamer formation. Since TIR-1b was used in our in vivo studies, we expressed and purified an MBP tagged ceSAMTIR expression construct as a mimic of TIR-1b and evaluated whether citrate activates this construct. As a control, we also evaluated the effect of citrate on the *C. elegans* TIR domain (ceTIR). Like the human TIR domain, the NAD^+^ hydrolase activity of both ceTIR and ceSAMTIR was greater in citrate, where ceTIR and ceSAMTIR activity were enhanced 22-fold and 1.4-fold respectively (Figure S5B). While citrate does not activate the ceSAMTIR construct to the same extent as the ceTIR domain alone, mutations that disrupt the TIR–ARM interface or deletion of the ARM domain activate SARM1 to a similar extent (i.e., 1.8– 2.4–fold) (Jiang et al., 2020).

Having demonstrated that ceSAMTIR activity is enhanced in citrate, we next sought to determine whether ceSAMTIR undergoes a phase transition. To do this, we incubated ceSAMTIR with increasing concentrations of citrate, centrifuged the samples, removed the soluble fraction, and resuspended the insoluble pelleted fraction. All fractions were analyzed by SDS-PAGE for the presence of ceSAMTIR. At citrate concentrations 250 mM and below, ceSAMTIR was predominantly located in the supernatant. By contrast, at citrate concentrations of 500 mM and above, the protein was increasingly found in the pellet, such that at 1000 mM citrate, ceSAMTIR was primarily located in the pellet.

Moreover, enzymatic activity in the fractions correlated with the location of ceSAMTIR (Figure S5C). These data are consistent with the hypothesis that longer versions of SARM1/TIR-1 also undergo a phase transition.

These findings have been included in the manuscript. The additional text reads:

“To this end, we opted to study the SARM1 orthologue, TIR–1 in the GABA motor nervous system of *C. elegans*. *C. elegans* encodes several TIR-1 splice variants including TIR-1b and TIR-1d which lack the ARM domain (Figure S5A ), but contain the tandem SAM domains essential for octamer formation. […] These data are consistent with the hypothesis that longer versions of SARM1/TIR-1 also undergo a phase transition that correlates with enzyme activation.”

We also edited the conclusions section by modifying an existing paragraph.

“Significant cellular and biochemical data supports the notion that the ARM domain interacts with and inhibits the TIR domain. […] Moreover, the data suggest that the ARM domain prevents TIR domain multimerization and limits full activation of the enzyme.”

Finally, we did tone down our final conclusions by replacing ‘…translates into an exponential increase in activity…’ to ‘…translates into increased activity …’

Reviewer #1 (Recommendations for the authors):A few key experiments, including the in vitro activation by PEG or citrate, and the puncta formation in HEK293T cells, should be replicated using the full length SARM1. If similar results are obtained with the full length SARM1, the conclusion will be more strongly supported and further enhance the connection to physiology/pathophysiology. Given that the in vivo data is obtained using *C. elegans*, the *C. elegans* full length SARM1 could be used for such study.

As described above in our response to essential revisions, we have demonstrated that citrate also activates TIR1b, a *C. elegans* isoform of SARM1 (see above). We believe that the puncta formation studies in HEK293T cells are beyond the scope of the manuscript, especially since we showed that puncta formation occurs in vivo. Additionally, constructs encoding full length human SARM1 are unstable due to their high GC content and this issue has presented a technical challenge to completing those studies. As such, we elected to proceed without those studies.

The observation that citrate promotes axon degeneration in C. elegans is interesting. The manuscript may use a little more discussion on this topic. For example, it would be interesting to discuss whether there are connections between citrate and axon degermation reported in the literature and whether citrate could be a physiological/pathophysiological trigger for axon degeneration in mammals.

We have added additional discussion of this phenomenon to the discussion. Reviewers 2 and 3 made a similar request. The additional text reads:

**“**Given that citrate is a mitochondrial metabolite, it is tempting to speculate that citrate, either from stressed mitochondria or from inhibition of the cytosolic enzyme ATP-citrate lyase, might promote a similar phase transition in vivo. […] Further research is needed to fully address this possibility.”

Reviewer #3 (Recommendations for the authors):This study of SARM1 builds on past information and extends it into a more complete picture of the conversion between inactive and active enzyme forms. A biological advance comes from the discovery that citrate can activate the enzyme. Citrate is then used in C. elegans to cause in vivo activation of the enzyme with subsequent neuronal damage. There is much to appreciate in this work as it advances the understanding of SARM1 activity and its role in neuronal degradation. There are some questions and areas of improvements to be made, summarized in the comments below. Once these are addressed, the manuscript should be published.An excellent TOC figure, it is worth more than the proverbial 1000 words.

We thank the reviewer for their kind words.

The title needs to include the catalytic function, perhaps "A Phase Transition Enhances the Catalytic Activity of SARM1, an NAD+ glycohydolase'.

We have changed the title to: “A Phase Transition Enhances the Catalytic Activity of SARM1, an NAD+ glycohydrolase involved in neurodegeneration”.

Figure 1 legend should explain proposed domain function.

We have modified the figure legend as requested. It now reads:

“Figure 1. SARM1 Reaction and Domain Structure. […] SARM1 consists of three domains: an autoinhibitory HEAT/armadillo (ARM) domain, two tandem sterile α motif (SAM) domains that promote octamerization, and a C–terminal toll interleukin receptor (TIR) domain that catalyzes NAD^+^ hydrolysis.”

Table 1 and the description "As before, we found that ZnCl2 inhibits SARM1 noncompetitively with a Ki value of 1.0 ± 0.1 μM." Do not appear to match."we found that berberine chloride inhibits SARM1 noncompetitively in the presence of PEG 3350 with a Ki value of 120 ± 10 μM." What is it? A Zn-chelator? Why is it being compared to Zn?

We apologize for the confusion. ZnCl2 and Berberine chloride are two separate inhibitors that were discovered by the Thompson lab. We evaluated their IC_50_ and mechanism of inhibition in the presence of PEG and compared those to previously established values obtained from lysates to establish that PEG was not influencing their inhibition properties, i.e. potency and mechanism of inhibition. We additionally performed similar experiments with pure protein in absence of PEG. We hope that this information clarifies the issue for the reviewer. To avoid confusion for other readers, we have noted in the manuscript that berberine chloride is a second independent inhibitor. The modified text reads:

**“**For a secondary confirmation that PEG does not impact the in vitro properties of SARM1, we tested a different inhibitor, berberine chloride that was previously identified from a high throughput screen (Loring et al., 2020b).”

Throughout, the units of uM/sec are ambiguous without knowing the enzyme concentration, requiring a search through the methods/legends. In contrast a rate in the traditional enzymatic units of turnovers / sec (simply s^-1^) seem more descriptive and are used in several critical places, eg. Figure 3 l. The reviewer is reluctant to ask the authors to make all the changes, but include as a note for future reports.

We thank the reviewer for the suggestion and will make use v/[E] values in future reports.

Even in its most active form SARM1 has a k_cat_/K_m_ around 103 s^-1^M^-1^s^-1^, making it a sleepy enzyme, the authors should compare with other known NAD glycohydrolases, and comment as to this difference and how much SARM1 there is in neuronal bodies (300 nM is cited) relative to an ability to deplete NAD pools.

We thank the reviewer for the suggestion and agree that the k_cat_/K_m_ values obtained for SARM1 (~2000 s^-1^M^-1^s^-1^) are not particularly robust, however, they are inline with other eukaryotic enzymes that we have studied including the protein arginine deiminases, protein arginine methyltransferases, and nicotinamide N-methyltransferase ^5-8^. Nonetheless, we have noted this issue in the results and suggested that the slow kinetics correlates with the fact that degeneration occurs on the hour time scale. Here is the additional text:

“While the *k*_cat_/*K*_m_ values obtained for SARM1 (~2000 s^-1^M^-1^s^-1^) are not particularly robust, they are in-line with other eukaryotic enzymes that we have studied e.g., the protein arginine deiminases, protein arginine methyltransferases, and nicotinamide Nmethyltransferase (Knuckley et al., 2007; Knuckley et al., 2010; Loring and Thompson, 2018; Osborne et al., 2007), and the slow kinetics likely relate to the fact that degeneration occurs on the hour time scale (Mack et al., 2001; Osterloh et al., 2012).”

“…cADPR inhibits SARM1 in the presence of PEG with an IC_50_ of 100 ± 10 μM and a competitive inhibition pattern…” What is the mechanism? If both cADPR and ADPR are competitive, it supports a rapid-equilibrium random release of products.

Respectfully, we previously showed that SARM1 hydrolyzes NAD+ via an ordered uni-bi reaction in which nicotinamide is released prior to ADPR ^4^. Since ADPR is released last, regenerating the free enzyme, competitive inhibition with respect to NAD+ is expected. cADPR is also expected to act as a competitive inhibitor because cADPR is a minor alternative product and not an obligate intermediate as demonstrated by our current product specificity ratio data and single turnover data. Moreover, this conclusion is consistent with our prior demonstration that ENAD is efficiently hydrolyzed by SARM1; the etheno bridge in ENAD blocks the N1 position, which was previously shown to be the site of cyclization ^4^. We hope that are explanation clarifies the issue for the reviewer.

The statement "Although the percentage of damaged axons in the citrate treated animals 2-3 h post-treatment is higher than untreated controls the data are not significant (Figure 6I)." is an obvious logical error. Not significant means not significant and the statement should be that there is no significant change in damaged axons at 2-3 h.

We apologize for the error. We have modified the sentence so that it now reads: “No significant change in the percentage of damaged axons was observed in the citrate treated animals 2–3 h post–treatment (Figure 6I).

The studies with C. elegans, citrate activation and axonal damage are a valuable addition. I was waiting for the NAD analysis to follow – can the authors measure NAD in control vs untreated C.e.?

Respectfully, it is well established the neuronal damage requires NAD+ hydrolysis in a SARM1 dependent manner. Given that we show that citrate induced neuronal damage is TIR1/SARM1 dependent the proposed experiments are of limited value and beyond the scope of the manuscript.

A comment needs to address the 500 mM citrate effects on SARM1 versus the 8 mM citrate effect on the C. elegans homologue of SARM1.

Reviewers 1 and 2 made a similar comment. Please see our detailed response to reviewer 1.

References:

1. Gerdts, J.; Brace, E. J.; Sasaki, Y.; DiAntonio, A.; Milbrandt, J., SARM1 activation triggers axon degeneration locally via NAD(+) destruction. Science 2015, 348 (6233), 453‐7.

2. Wang, J. T.; Medress, Z. A.; Vargas, M. E.; Barres, B. A., Local axonal protection by WldS as revealed by conditional regulation of protein stability. Proc Natl Acad Sci U S A 2015, 112 (33), 10093100.

3. Essuman, K.; Summers, D. W.; Sasaki, Y.; Mao, X.; DiAntonio, A.; Milbrandt, J., The SARM1

Toll/Interleukin‐1 Receptor Domain Possesses Intrinsic NAD(+) Cleavage Activity that Promotes

Pathological Axonal Degeneration. Neuron 2017, 93 (6), 1334‐1343 e5.

4. Loring, H. S.; Icso, J. D.; Nemmara, V. V.; Thompson, P. R., Initial Kinetic Characterization of

Sterile Alpha and Toll/Interleukin Receptor Motif‐Containing Protein 1. Biochemistry 2020, 59 (8), 933942.

5. Knuckley, B.; Bhatia, M.; Thompson, P. R., Protein arginine deiminase 4: evidence for a reverse protonation mechanism. Biochemistry 2007, 46 (22), 6578‐6587.

6. Knuckley, B.; Causey, C. P.; Jones, J. E.; Bhatia, M.; Dreyton, C. J.; Osborne, T. C.; Takahara, H.; Thompson, P. R., Substrate specificity and kinetic studies of PADs 1, 3, and 4 identify potent and selective inhibitors of protein arginine deiminase 3. Biochemistry 2010, 49 (23), 4852‐4863.

7. Osborne, T. C.; Obianyo, O.; Zhang, X.; Cheng, X.; Thompson, P. R., Protein arginine methyltransferase 1: positively charged residues in substrate peptides distal to the site of methylation are important for substrate binding and catalysis. Biochemistry 2007, 46 (46), 13370‐81.

8. Loring, H. S.; Thompson, P. R., Kinetic Mechanism of Nicotinamide N‐Methyltransferase. Biochemistry 2018, 57 (38), 5524‐5532.